# Delayed tumor-draining lymph node irradiation preserves the efficacy of combined radiotherapy and immune checkpoint blockade in models of metastatic disease

Irma Telarovic [1], Carmen S. M. Yong[1,2], Lisa Kurz[3], Irene Vetrugno [1], Sabrina Reichl[1,4], Alba Sanchez Fernandez[1], Hung-Wei Cheng [3], Rona Winkler[1], Matthias Guckenberger [4], Anja Kipar [5], Burkhard Ludewig [3] & Martin Pruschy [1] ✉

Cancer resistance to immune checkpoint inhibitors motivated investigations into leveraging the immunostimulatory properties of radiotherapy to overcome immune evasion and to improve treatment response. However, clinical benefits of radiotherapy-immunotherapy combinations have been modest. Routine concomitant tumor-draining lymph node irradiation (DLN IR) might be the culprit. As crucial sites for generating anti-tumor immunity, DLNs are indispensable for the in situ vaccination effect of radiotherapy. Simultaneously, DLN sparing is often not feasible due to metastatic spread. Using murine models of metastatic disease in female mice, here we demonstrate that delayed (adjuvant), but not neoadjuvant, DLN IR overcomes the detrimental effect of concomitant DLN IR on the efficacy of radio-immunotherapy. Moreover, we identify IR-induced disruption of the CCR7-CCL19/CCL21 homing axis as a key mechanism for the detrimental effect of DLN IR. Our study proposes delayed DLN IR as a strategy to maximize the efficacy of radio-immunotherapy across different tumor types and disease stages.

The introduction of ICIs redefined the landscape of modern cancer treatment by inducing unprecedented, durable responses in patients with metastatic, treatment-resistant disease[1]. However, up to 80% of patients eligible for ICIs according to the current criteria experience either a primary non-response or develop resistance over the course of treatment[2,3]. In order to fully realize the potential of immunotherapy, it is thus necessary to deepen our understanding of the factors dictating

the response to ICIs and subsequently optimize the integration of ICIs into current treatment protocols.

The discovery of immunostimulatory properties of irradiation (IR) has shifted the view of radiotherapy as a purely directly cytotoxic, local treatment modality. Tumor IR has been shown to initiate a complex cascade of events that includes the release of tumor antigens and danger-associated molecular patterns (DAMPs) as part of

[1]Laboratory for Applied Radiobiology, Department of Radiation Oncology, University Hospital Zurich, University of Zurich, Zurich, Switzerland. [2]Department of Immunology, University Hospital Zurich, University of Zurich, Zurich, Switzerland. [3]Institute of Immunobiology, Medical Research Center, Kantonsspital St. Gallen, St. Gallen, Switzerland. [4]Department of Radiation Oncology, University Hospital Zurich, University of Zurich, Zurich, Switzerland. [5]Laboratory for Animal Model Pathology, Institute of Veterinary Pathology, Vetsuisse Faculty, University of Zurich, Zurich, Switzerland. ✉e-mail: martin.pruschy@uzh.ch

immunogenic cell death, upregulation of MHC class I, death receptors and co-stimulatory molecules, and production and release of proinflammatory cytokines and chemokines[4,5]. These changes in the tumor microenvironment collectively support cross-priming of tumor-directed CD8+ T cells and simultaneously increase the vulnerability of tumor cells to the immune system, thus ultimately inducing both local and systemic immune-mediated tumor cell killing[6,7]. This propensity of IR to act as an in situ cancer vaccine has therefore redefined radiotherapy as an important contributor to the cancer-immunity cycle[8]. In line with these findings, a combinatorial approach with immunotherapy has been proposed in the early 2000s[9] as a strategy to harness the potential synergism between the two treatment modalities. Preclinical studies have since provided promising results and strong mechanistic evidence favoring the combined treatment[4,10–12], paving the way towards the clinical introduction of radio-immunotherapy. However, the efficacy on the clinical level has so far been disappointing[13].

Common practice of tumor-draining lymph node (DLN) co-irradiation has been suggested as one of the possible culprits for the modest clinical benefit of radioimmunotherapy[13–15]. On the one hand, DLNs are a common site of early metastatic spread and therefore often irradiated as part of curative radiotherapy treatment. The rationale for this approach has been justified in multiple clinical trials and over different solid cancer types, with undisputable benefits established in e.g. breast cancer[16], prostate cancer[17] and head and neck squamous cell carcinoma[18]. On the other hand, DLNs are a crucial component of the cancer-immunity cycle[8,15]. Following the uptake of tumor antigens, activated tumor-patrolling dendritic cells (DCs) migrate through the lymphatic system towards the DLNs, which foster a unique environment capable of supporting the priming of naïve tumor-antigen-specific T cells. Therefore, therapeutic sterilization of DLNs at the time of tumor IR might abrogate the immunostimulatory effects of IR, by rendering the DLNs dysfunctional at a critical timepoint. Indeed, emerging preclinical evidence suggest that DLNs may be pivotal for the successful development of anti-tumor immunity in response to both radiotherapy and immunotherapy[19–23]. Lymphatic sparing has therefore been evaluated preclinically as a strategy to improve the therapeutic response for a subset of patients with clinically negative lymph nodes, whose DLNs are currently routinely therapeutically sterilized as part of elective nodal IR[24–26]. However, for most patients with nodal involvement or with a high risk for microscopic involvement, lymphatic sparing is not a viable option, as DLN IR in this setting is a major contributor to disease control[16–18].

In this work, we propose a treatment strategy based on temporal distancing between IR of the tumor and IR of DLNs to maximize the positive effects of tumor IR on the anti-tumor immune response, while simultaneously preserving the beneficial effect of DLN IR on metastatic tumor cell killing. To investigate this approach, we develop a murine model of metastatic disease using mice bearing tumors with an early disease spread into the DLNs and mice with bilateral tumors. Using a state-of-the-art small animal image-guided radiotherapy platform, we develop a protocol for high-precision IR that enables us to identify and precisely include or exclude the DLNs from the treatment field. Based on an extensive kinetics study of IR-induced changes in the immuno-phenotype of the tumor and the DLNs, we define timepoints for early (neoadjuvant) and delayed (adjuvant) DLN IR relative to tumor IR. In a series of investigations on the level of mechanism and efficacy, we identify the optimal treatment scheme to still irradiate the DLNs and to overcome the observed detrimental effect of conventional (concomitant) DLN IR. Furthermore, we investigate mechanistic aspects of the elusive communication between the primary tumor and the DLNs upon IR. Taken together, our results indicate that a rationally designed delay of DLN IR is a highly promising and easy way to maintain this communication for as long as necessary and to implement a strategy with the potential to substantially improve the

response to combined radioimmunotherapy across different tumor types and disease stages.

## Results

### Accurate lymph node targeting using image-guided radiotherapy in a murine model of metastatic disease

To model the clinical setting of nodal involvement at the time of treatment, we developed a murine model of melanoma with an early spread into the DLNs. A luciferase-expressing B16F10 mouse melanoma cell line (B16F10-Luc) was used to enable bioluminescence-based evaluation of the presence of tumor cells in the DLNs. On day 6 after tumor cell injection (when tumors reached an average size of 80 mm³; henceforth defined as day 0 relative to tumor IR), 80% of the mice were positive for tumor cells in the axillary, brachial and/or inguinal lymph nodes (Fig. 1A–C), which we previously identified as the DLNs of this region using Evans Blue dye[27].

Using a small-animal image-guided radiotherapy platform, all DLNs could reliably be visualized (Fig. 1D) and subsequently irradiated or spared (Fig. 1E–G). Tumor IR was performed using a rectangular 8 × 12 mm field (Fig. 1E, left). When indicated, DLN IR was performed using two additional rectangular 8 × 12 or circular 5 mm fields (depending on the individual mouse anatomy) to accurately target the two DLNs (Fig. 1E, right).

In a previous study, we developed the framework and quantitatively verified the feasibility of such high-precision, volume-oriented small animal radiotherapy treatment[28]. For illustrative purposes, the accuracy of treatment planning and execution is depicted here by the pattern of IR-induced depigmentation which precisely corresponded to the treatment plan (Fig. 1F) and using immunohistochemistry on sections of axillary, brachial and inguinal DLNs and contralateral non-DLNs (NDLNs) to detect phosphorylated histone H2AX (γH2AX) as a biomarker for exposure to IR (Fig. 1G)[29]. In a visual comparison between the representative inguinal lymph node sections of the three mice treated with three different treatment approaches (CT imaging only; CT imaging and tumor IR; CT imaging, tumor and DLN IR), the highest degree of γH2AX positivity could clearly be observed in the irradiated DLN (Fig. 1G, top right panel). As expected, DLNs of the two mice treated with either CT imaging or tumor-only IR showed only a low level of γH2AX positivity, comparable to their respective sham-irradiated NDLNs (Fig. 1G, left and middle panel). Interestingly, a modest increase in γH2AX positivity of the sham-irradiated NDLNs could be visually correlated with an increase in the irradiated volume. We previously demonstrated that even very small, tumor-only directed treatment fields significantly affect circulating lymphocytes[27]. Thus, the apparent increase in γH2AX positivity of sham-irradiated NDLNs could be due to IR-induced damage in the circulating lymphocytes which pass through the treatment fields during IR.

### Concomitant draining lymph node irradiation does not affect the tumor response to radiotherapy alone

In current clinical practice, lymph nodes infiltrated with metastases are irradiated at the same time (concomitantly) as the tumor. Therefore, we first sought to determine the importance of conventional, concomitant DLN IR in the setting of radiotherapy alone.

B16F10-Luc mouse melanoma tumors were developed as described above. Tumor growth was followed over 50 days in response to tumor-only IR ("TM IR" group) and the combination of tumor IR and concomitant IR of the axillary, brachial and inguinal lymph nodes which drain the tumor site ("TM + C-DLN IR" group), with each individual target receiving a single high dose of 15 Gy (Fig. 2A). Both radiotherapy treatment regimens induced an extended tumor growth delay compared to non-irradiated mice ("Sham IR" group) (Fig. 2B–E). However, no significant differences were observed in the tumor response between "TM IR" and "TM + C-DLN IR" groups, as demonstrated by the Kaplan–Meier analysis whereby the median time to

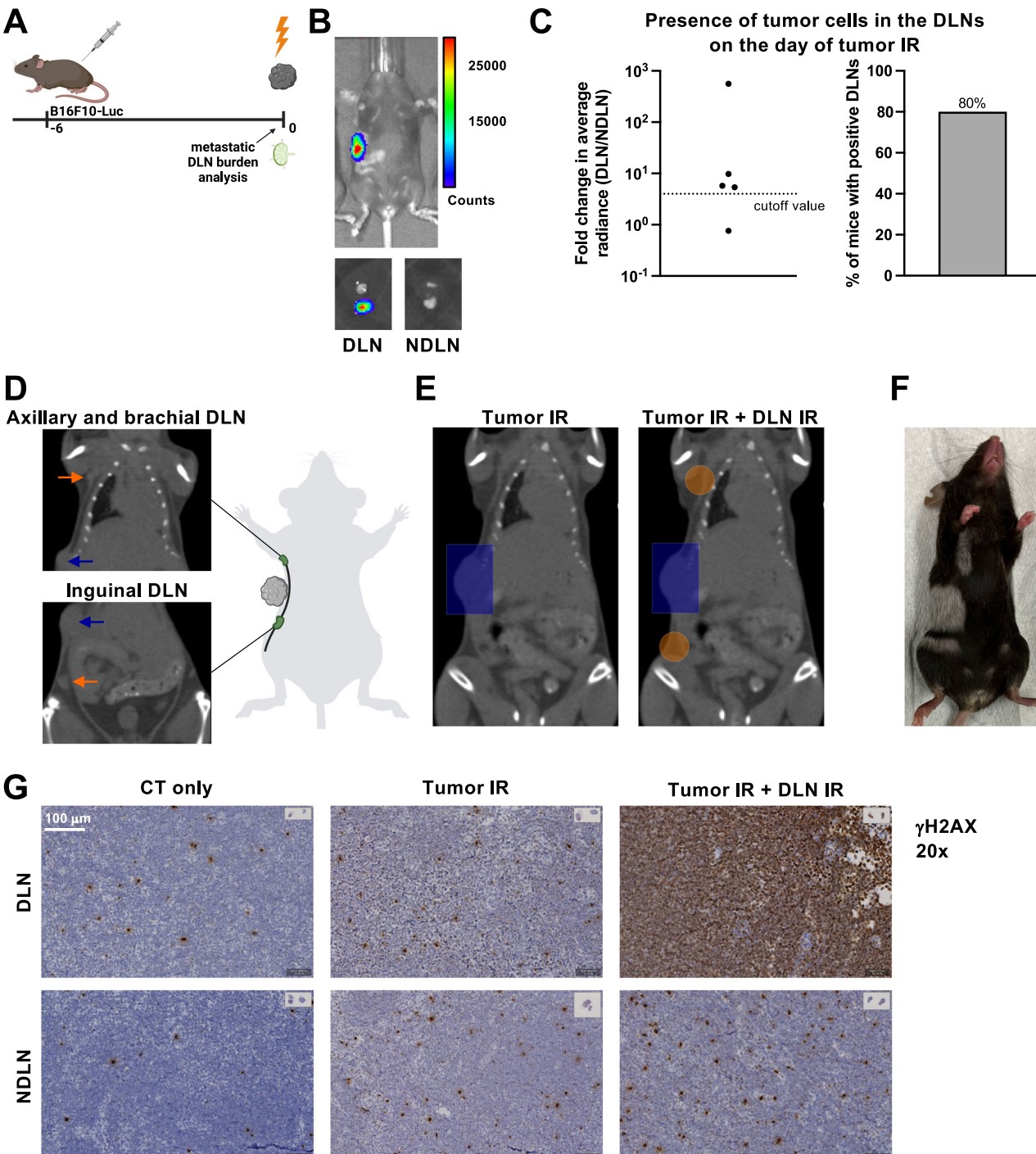

**Fig. 1 | Accurate lymph node targeting using image-guided radiotherapy in a murine model of metastatic disease. A** A luciferase-expressing B16F10 mouse melanoma cell line was used to generate a metastatic melanoma tumor model. The day of tumor IR (day 0) was defined as the day when tumors reached an average size of 80 mm³ (day 6 after cell injection). **B** Top: Representative image of a tumor-bearing mouse on the day of tumor IR. Bottom: Excised lymph nodes from the same mouse, DLNs on the left and contralateral NDLNs on the right, with the axillary lymph node on top and the inguinal lymph node on the bottom of each image. **C** Quantitative analysis of the presence of tumor cells in the DLNs on the day of tumor IR. Left: Fold change in the average radiance of the DLN compared to the contralateral NDLN. Dotted line indicates the cutoff value for tumor cell positivity, defined as a 300% increase in the signal over the NDLN (fold change >4). Each dot represents an individual mouse. Right: Quantitative representation of the mice with tumor cell positive DLNs on the day of tumor IR, using the cutoff value defined

above. *n* = 5 mice. **D** CT image on the day of tumor IR. Orange arrows point to the axillary and brachial (top), and inguinal DLNs (bottom). Blue arrow indicates the tumor. **E** Radiotherapy treatment plans. Left: Tumor-only IR is performed using a rectangular 8 × 12 mm field, shown in blue. Right: Two additional circular 5 mm fields (in orange) are used to target the DLNs. **F** Mouse with depigmentation corresponding to area of skin exposed to IR during tumor and DLN IR, photographed on day 60 after delivering 15 Gy. **G** γH2AX staining performed 30 min after delivering 15 Gy. Top row: DLN, bottom row: contralateral NDLN. Left column: planning CT only, middle column: planning CT and tumor IR, and right column: planning CT, tumor IR and DLN IR. Representative sections from *n* = 1 mouse per treatment group, 6 lymph nodes per mouse. Source data are provided as a Source Data file. Figures **A** and **D**, created with BioRender.com, released under a Creative Commons Attribution-NonCommercial-NoDerivs 4.0 International license.

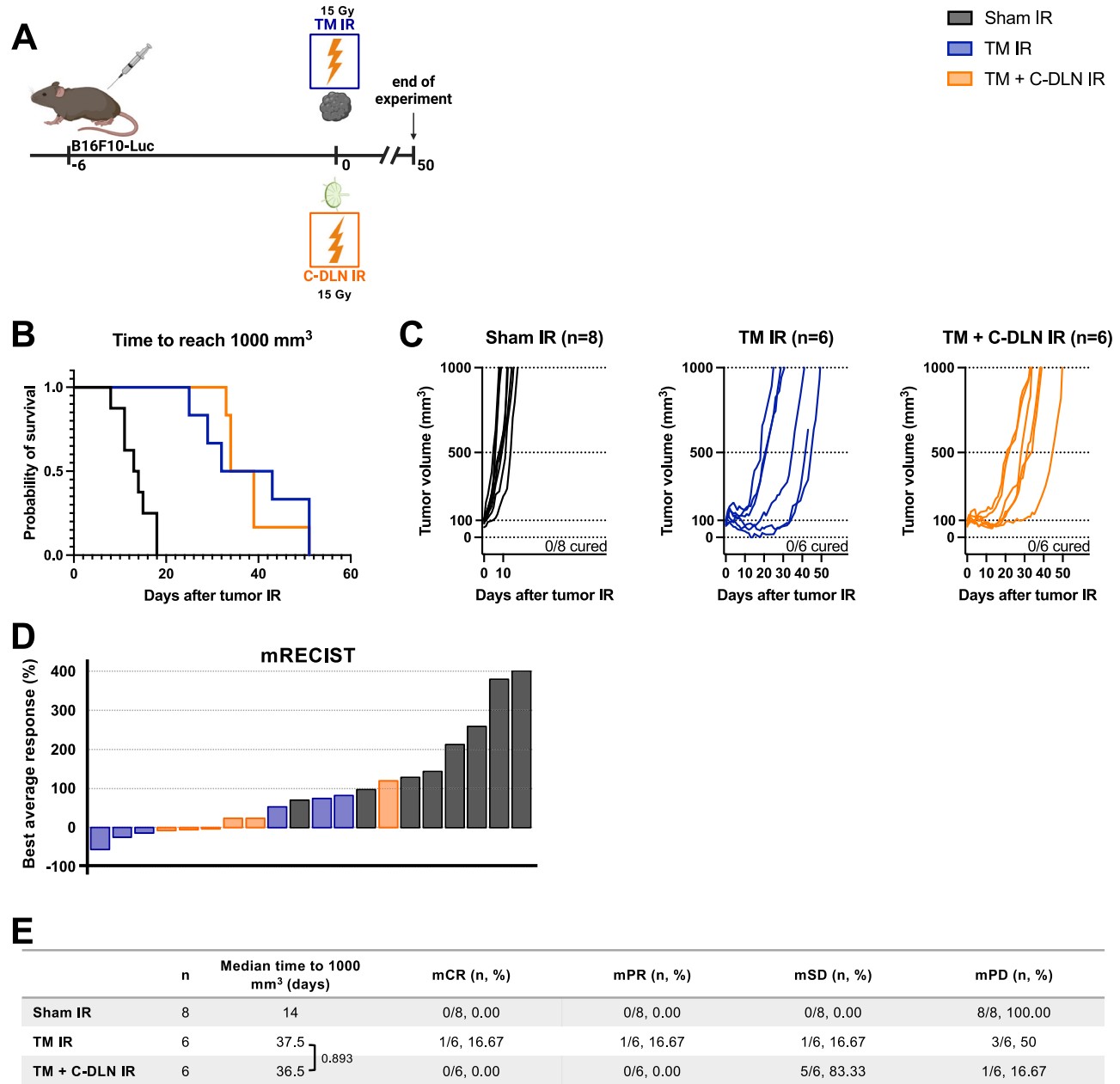

**Fig. 2 | Concomitant draining lymph node irradiation does not affect the tumor response to radiotherapy alone. A** B16F10-Luc tumor-bearing mice received tumor IR, with or without concomitant DLN IR. "Sham IR" group (black) was sham-irradiated, "TM IR" group (blue) received tumor IR and "TM + C-DLN IR" group (orange) received DLN IR concomitantly to tumor IR. Tumor growth was followed over 50 days. **B–E** Treatment response represented by Kaplan–Meier survival analysis (**B** and **E**), individual tumor growth curves (**C**) and a waterfall plot derived from the mRECIST analysis (**D**). Time to reach 1000 mm³ was used as the endpoint for Kaplan–Meier analysis. Each line in **C** and each bar in **D** represents an individual mouse. Parameters derived from the mRECIST analysis in **D** and **E** are described in the Methods section. mCR, complete response; mPR, partial response; mSD, stable disease; mPD, progressive disease. Number of mice in each group is indicated in the corresponding graph title in **C**. Logrank test (Mantel–Cox) was used to compare the survival curves; corresponding *p* values are displayed in **E**. All *p* values are displayed, with *, ** and *** indicating $p < 0.05$, $p < 0.01$ and $p < 0.001$, respectively. Source data are provided as a Source Data file. Figure **A**, created with BioRender.com, released under a Creative Commons Attribution-NonCommercial-NoDerivs 4.0 International license.

reach 1000 mm³ was 14, 43 and 36.5 days for "Sham IR", "TM IR" and "TM + C-DLN IR" group, respectively (Fig. 2B, E). Likewise, modified Response Evaluation Criteria in Solid Tumors (mRECIST) analysis did not indicate differences between the two irradiated groups (Fig. 2D, E, see "Methods" section for details on the analysis).

Taken together, these data demonstrate that concomitant DLN IR does not alter the treatment response in the setting of radiotherapy alone.

## Concomitant draining lymph node irradiation abrogates the beneficial effect of radioimmunotherapy

We and others have previously demonstrated in different tumor models that DLN IR impacts the treatment response in the context of combined radioimmunotherapy[24,27]. Therefore, we used the experimental setup described above, but now using combination therapy with an immunomodulator instead of radiotherapy alone. We used the clinically approved ICI α-CTLA-4, which boosts anti-tumor immunity

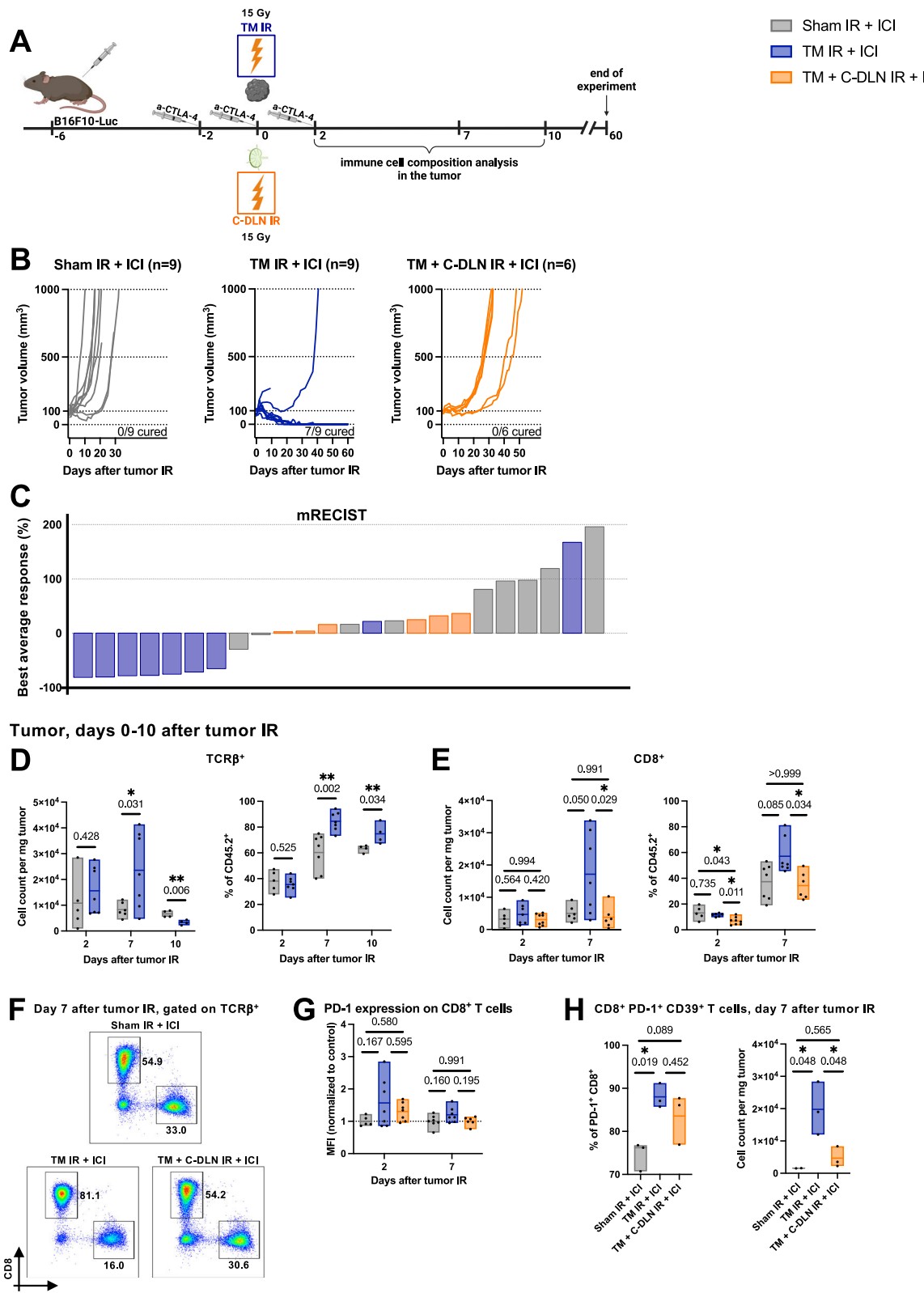

Tumor, days 0-10 after tumor IR

primarily by supporting the immune priming phase in the lymph node[4].

B16F10-Luc tumor-bearing mice received immunotherapy on days −2, 0 and 2, relative to tumor IR (Fig. 3A) and tumors were irradiated with or without concomitant DLN IR ("TM + C-DLN IR + ICI" and "TM IR + ICI" groups, respectively). Tumor response on the level of treatment efficacy was evaluated over 60 days (Fig. 3B, C) and the tumor immune microenvironment within 10 days after radiotherapy (Fig. 3D–H).

The addition of immunotherapy drastically improved the response to tumor-only IR, with 7/9 mice achieving complete and durable regression (Fig. 3B, "TM IR + ICI" group), as opposed to 0/6 mice treated with radiotherapy alone (Fig. 2C, "TM IR" group) and 0/9 mice treated with ICI alone (Fig. 3B, "Sham IR + ICI" group). In contrast, all mice treated with combined radioimmunotherapy and concomitant DLN IR progressed after a short period of growth stagnation (Fig. 3B, "TM + C-DLN IR + ICI" group), similar to their

**Fig. 3 | Concomitant draining lymph node irradiation abrogates the beneficial effect of radioimmunotherapy. A** B16F10-Luc tumor-bearing mice received α-CTLA-4 and tumor IR, with or without concomitant DLN IR. All groups received α-CTLA-4. "Sham IR + ICI" group (gray) was sham-irradiated, "TM IR + ICI" group (blue) received tumor IR, and "TM + C-DLN IR + ICI" group (orange) received DLN IR concomitantly to the tumor IR. Tumor growth was followed over 60 days. Immune cell composition was analyzed at different timepoints, as indicated. Treatment response represented by individual tumor growth curves (**B**) and a waterfall plot derived from the mRECIST analysis (**C**). Each line in **B** and each bar in **C** represents an individual mouse. Best average response value in **C** is described in the Methods section. Number of mice is indicated in **B**. **D**–**H** Tumor immune microenvironment. Gating strategy is shown in Supplementary Fig. 1A. Each dot represents an individual mouse. Floating bars span from the minimal to the maximal value of each group. Line indicates the mean. **D** Tumor-infiltrating T cells (TCRβ⁺). Left: cell count per mg tumor, right: TCRβ⁺ cells as a percent of CD45⁺ cells. **E** Tumor-infiltrating CD8⁺ T cells. Left: cell count per mg tumor, right: CD8⁺ cells as a percent of CD45⁺

cells. **F** Representative plots on day 7 after tumor IR. Numbers indicate the percentages of CD8⁺ and CD4⁺ T cells within the T cell compartment. **G** PD-1 expression on CD8⁺ T cells, expressed as the geometric mean of the fluorescence intensity (MFI), normalized to the average MFI value of the "Sham IR + ICI" group. **H** CD8⁺ PD-1⁺ CD39⁺ T cells. Left: expressed as a percent of all PD-1⁺ CD8⁺ T cells, right: CD8⁺ PD-1⁺ CD39⁺ T cell count per mg tumor. $n \geq 4$ for **D**, $n \geq 5$ for **E** and **G**, and $n = 3$ mice per group for **H** (exact numbers provided in Source Data file). Data were tested for normality using the Shapiro–Wilk test. For data following a normal distribution, treatment groups were compared using the two-sided unpaired $t$ test (**D**) or one-way ANOVA with Holm–Sidak's multiple comparisons test (**E**, left, **G** and **H**). For non-normally distributed data, the comparison was performed using the Kruskal–Wallis test with Dunn's multiple comparisons test (**E**, right). All $p$ values are displayed, with *, ** and *** indicating $p < 0.05$, $p < 0.01$ and $p < 0.001$, respectively. Source data are provided as a Source Data file. Figure **A**, created with BioRender.com, released under a Creative Commons Attribution-NonCommercial-NoDerivs 4.0 International license.

counterpart in the setting of radiotherapy alone (Fig. 2B, "TM + C-DLN IR" group).

We next performed a kinetics study of the tumor immunophenotype to characterize immune cell infiltration in response to tumor IR, with or without concomitant DLN IR (Fig. 3D–F and Supplementary Fig. 1A–C). In comparison to the "Sham IR + ICI" group, a significant increase in the absolute number of tumor-infiltrating immune cells (CD45⁺) in the tumors of mice treated with "TM IR + ICI" was observed already on day 2 after IR (Supplementary Fig. 1B). On day 7 after tumor IR, the immune infiltrate became T cell dominant (Fig. 3D), as evidenced by the absolute number (Fig. 3D, left) and the proportion of TCRβ⁺ cells in the tumor (Fig. 3D, right). On day 10 after tumor IR, the immune infiltrate remained dominated by T cells, however the initial increase in the absolute number of T cells was not present anymore. Having identified day 7 as the peak of IR-induced T cell infiltration in our model, we performed a more detailed analysis of the tumor immune cell composition over 7 days after tumor IR and compared T cell subpopulations in tumors of mice treated with "Sham IR + ICI" (gray bar), "TM IR + ICI" (blue bar) and "TM + C-DLN IR + ICI" (orange bar) (Fig. 3E, F). T cell infiltration in response to tumor IR could thereby be attributed to cytotoxic CD8⁺ T cells, which substantially increased both in absolute numbers (Fig. 3E, left) and in terms of the percentage within the CD45⁺ compartment (Fig. 3E, right). Importantly, the shift towards a CD8⁺-dominated tumor microenvironment could only be detected in the "TM IR + ICI" group. In contrast, the immune infiltrate of mice that received concomitant DLN IR comprised a smaller proportion of CD8⁺ T cells on day 2 after tumor IR as compared to the other treatment groups. On day 7 after tumor IR, no significant differences within the CD8⁺ T cell compartment between the "Sham IR + ICI" and "TM + C-DLN IR + ICI" groups were evident. Notably, along with an increase in CD8⁺ T cells, the absolute number of CD4⁺ FOXP3⁺ regulatory T cells was also elevated on day 7 after IR in the "TM IR + ICI" group as compared to the other treatment groups (Supplementary Fig. 1C, left). However, this increase was not reflected in the proportion of regulatory T cells within the CD45⁺ compartment (Supplementary Fig. 1C, right).

To gain more insight into the phenotype of the tumor-infiltrating CD8⁺ T cells, we quantified the expression of the co-inhibitory receptors PD-1 and TIM-3 (Fig. 3G and Supplementary Fig. 1D). Compared to the "Sham IR + ICI+ and "TM + C-DLN IR + ICI" groups, a trend towards a higher expression of PD-1 and TIM-3 could be observed on tumor-infiltrating CD8⁺ T cells isolated from mice in the "TM IR + ICI" group both on day 2 and day 7 after tumor IR. Depending on the context, the expression of co-inhibitory receptors on tumor-infiltrating CD8⁺ T cells indicates either recent activation of antigen-specific cells or an exhausted state[30]. Therefore, we quantified the expression of ectonucleotidase CD39 on PD-1⁺ CD8⁺ T cells, which has recently been described as a specific marker of tumor-reactive, antigen-specific

tumor-infiltrating CD8⁺ T cells[31–33]. Indeed, in line with the effects observed on the level of efficacy, the majority of CD8⁺ PD1⁺ T cells within the "TM IR + ICI" group were positive for CD39, indicating an ICI/IR-facilitated antigen-specific immune response, but detectable only in this group whereby the DLN were not irradiated.

Collectively, these findings demonstrate that concomitant DLN IR abrogates the beneficial effect of radioimmunotherapy on the level of the tumor response and immune cell infiltration into the tumor microenvironment.

## Delayed (adjuvant) draining lymph node irradiation preserves the efficacy of radioimmunotherapy

The necessity of DLN IR in the setting of a metastatic disease with nodal involvement conflicts with the requirement of a functionally intact DLN for radioimmunotherapy. We hypothesized that delayed ("adjuvant") DLN IR relative to tumor IR would allow for the optimal development of anti-tumor immunity in response to combined radioimmunotherapy, while still preserving the benefits of DLN IR.

Based on the immune infiltration kinetics study described above, we probed a delay of 2 days ("A2-DLN IR") and 7 days ("A7-DLN IR") as adjuvant treatment schedules (Fig. 4A). Tumor response was evaluated on the level of treatment efficacy over 60 days (Fig. 4B–F) and immune cell infiltration at day 7 after tumor IR (Fig. 4G). As demonstrated by Kaplan–Meier survival analysis (Fig. 4B), area under the curve analysis (AUC) (Fig. 4C) and mRECIST analysis (Fig. 4E, F), the 7-day delay of DLN IR completely preserved the benefit of the combined treatment. Surprisingly, with a complete response (mCR) rate of 38.46%, even as little as a 2-day delay of DLN IR was superior to concomitant DLN IR (mCR of 0%). However, in comparison to the 7-day delay and tumor-only IR (mCR of 75% and 77.78%, respectively), "TM + A2-DLN + ICI" treatment schedule only partially preserved the efficacy of the combined treatment (Fig. 4F), thus suggesting only partial preservation of the DLN-dependent immune response.

To assess whether changes in the efficacies of the different treatment regimens correlate to differences in the immune cell composition of the tumor after treatment, we quantified the amount of tumor-infiltrating effector T cells (CD8⁺ CD44⁺) on day 7 after tumor IR (Fig. 4G). To be noted is that the DLNs have not been irradiated yet in the "TM + A7-DLN IR + ICI" treatment group at this timepoint, and therefore, the immune cell composition of the tumor corresponds to that of the "TM IR + ICI" group. The tumor microenvironment was dominated by T cell effectors in the "TM IR + ICI" (and thus also "TM + A7-DLN IR + ICI") and "TM + A2-DLN IR + ICI" treatment groups, with these cells comprising 55.55 ± 14.17% and 58.93 ± 4.48%, respectively, of the total CD45⁺ cell population in the tumor, which was substantially more than in the "Sham IR" group (14.27 ± 11.58%) (Fig. 4G, left). In contrast and in line with the tumor growth analysis, the "TM + C-DLN IR + ICI" regimen failed to induce this shift towards an

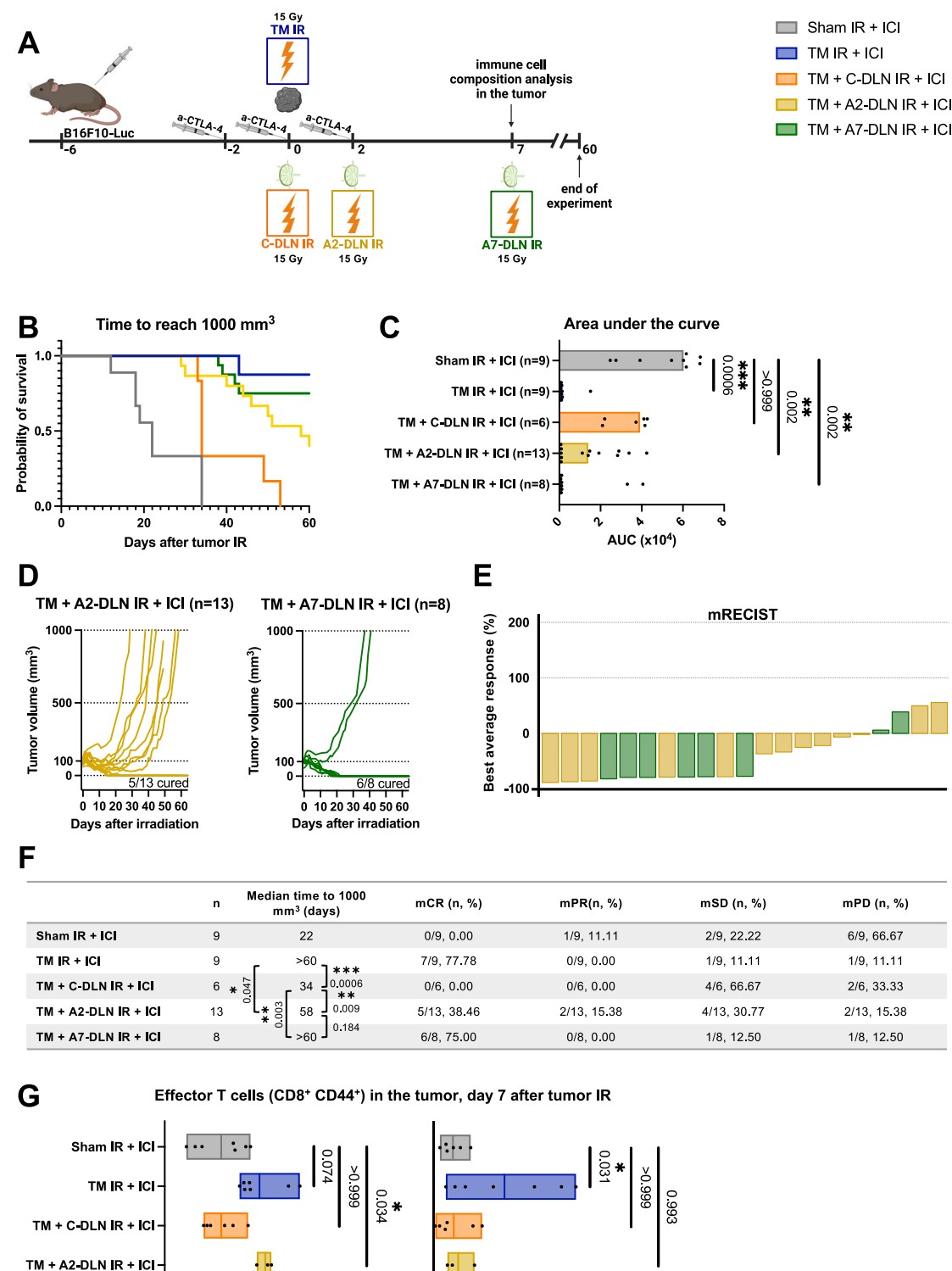

**F**

| | n | Median time to 1000 mm³ (days) | mCR (n, %) | mPR(n, %) | mSD (n, %) | mPD (n, %) |
|---|---|---|---|---|---|---|
| Sham IR + ICI | 9 | 22 | 0/9, 0.00 | 1/9, 11.11 | 2/9, 22.22 | 6/9, 66.67 |
| TM IR + ICI | 9 | >60 | 7/9, 77.78 | 0/9, 0.00 | 1/9, 11.11 | 1/9, 11.11 |
| TM + C-DLN IR + ICI | 6 | 34 | 0/6, 0.00 | 0/6, 0.00 | 4/6, 66.67 | 2/6, 33.33 |
| TM + A2-DLN IR + ICI | 13 | 58 | 5/13, 38.46 | 2/13, 15.38 | 4/13, 30.77 | 2/13, 15.38 |
| TM + A7-DLN IR + ICI | 8 | >60 | 6/8, 75.00 | 0/8, 0.00 | 1/8, 12.50 | 1/8, 12.50 |

effector-dominated microenvironment, with only 33.09 ± 4.48% of the immune cells classified as T cell effectors. Interestingly, in the comparison of absolute cell counts per mg tumor, a significant increase in the number of effector T cells could only be observed in the "TM IR + ICI"/"TM + A7-DLN IR + ICI" treatment group, and not in the tumors of mice treated with a 2-day delay of DLN IR (Fig. 4G, bottom).

In summary, the efficacy of radioimmunotherapy was successfully preserved in the setting of a metastatic disease with nodal involvement by delaying DLN IR. Differences in the tumor growth between the treatment groups correlated with a shift towards an effector T cell-dominated tumor microenvironment, which occurred only when the DLNs were temporarily or completely spared.

## Neoadjuvant draining lymph node irradiation fails to preserve the efficacy of radioimmunotherapy

Lymphopenia in response to localized radiotherapy may be quickly reversed, most probably due to rapid influx of lymphocytes from the non-irradiated compartments[25,27,34]. DLN IR prior to tumor IR

**Fig. 4 | Delayed (adjuvant) draining lymph node irradiation preserves the efficacy of radioimmunotherapy. A** B16F10-Luc tumor-bearing mice received α-CTLA-4 and tumor IR, with or without DLN IR at different timepoints. All groups received α-CTLA-4. "Sham IR + ICI" group (gray) was sham-irradiated, "TM IR + ICI" group (blue) received tumor IR, "TM + C-DLN IR + ICI" group (orange) received DLN IR concomitantly to the tumor IR, "TM + A2-DLN IR + ICI" (yellow) and "TM + A7-DLN IR + ICI" (green) received DLN IR delayed by 2 and 7 days, respectively. Tumor growth was followed over 60 days. Immune cell composition was analyzed on day 7 after tumor IR. **B–F** Treatment response represented by the Kaplan–Meier survival analysis (**B** and **F**), area under the curve (AUC) analysis (**C**), individual tumor growth curves (**D**) and a waterfall plot derived from the mRECIST analysis (**E**). Time to reach 1000 mm³ was used as the endpoint for Kaplan–Meier analysis. Each dot in **C**, each line in **D** and each bar in **E** represents an individual mouse. Bar width in **C** represents the median value of the corresponding group. Parameters derived from the mRE-CIST analysis in **E** and **F** are described in the Methods section. mCR complete response, mPR partial response, mSD stable disease, mPD progressive disease.

Number of mice in each group is indicated in **F**. **G** Tumor-infiltrating effector T cells on day 7 after tumor IR. Left: Effector T cells as a percentage of CD45$^+$ cells, right: cell count per mg tumor. Gating strategy is shown in Supplementary Fig. 1A. Each dot represents an individual mouse. Floating bars span from the minimal to the maximal value of each group. Line indicates the mean. $n \geq 3$ mice per group (exact numbers provided in Source Data file). Data were tested for normality using the Shapiro–Wilk test. For data following a normal distribution, treatment groups were compared using the one-way ANOVA with Holm–Sidak's multiple comparisons test (**G**, right). For non-normally distributed data, the comparisons were performed using the Kruskal–Wallis test with Dunn's multiple comparisons test (**C** and **G**, left). Logrank test (Mantel–Cox) was used to compare the survival curves; corresponding $p$ values are displayed in **F**. All $p$ values are displayed, with *, ** and *** indicating $p < 0.05$, $p < 0.01$ and $p < 0.001$, respectively. Source data are provided as a Source Data file. Figure **A**, created with BioRender.com, released under a Creative Commons Attribution-NonCommercial-NoDerivs 4.0 International license.

("neoadjuvant") may therefore allow for functional reconstitution of the DLNs at the time of tumor IR and even offer additional benefits, as extensive and prolonged invasion of metastatic cells in the DLN could lead to the development of an immunosuppressive microenvironment and thereby render the DLNs dysfunctional[15]. Thus, neoadjuvant DLN IR may kill the metastatic tumor cells, deplete the immunosuppression-promoting cells, and enable repopulation of the irradiated DLN with healthy cells capable of mounting an anti-tumor immune response.

We therefore evaluated the treatment efficacy in response to neoadjuvant DLN IR. Mice bearing B16F10-Luc tumors received tumor IR on day 9 after tumor cell injection (henceforth referred to as day 0), whereby the average tumor size was 115 mm³. DLN IR was either performed concomitantly (on day 0) or in a neoadjuvant setting (on day -7 relative to tumor IR), while immunotherapy was given as previously on days −2, 0 and 2, relative to tumor IR (Fig. 5A). As previously observed (Fig. 3), concomitant DLN IR abrogated the beneficial effect of radioimmunotherapy, as reflected in the Kaplan–Meier survival analysis and the mCR rate of 14.28% compared to the mCR rate of 50% in the "TM IR + ICI" group (Fig. 5B–E). Interestingly, neoadjuvant DLN IR completely failed to reverse the negative effect of concomitant DLN IR. With an mCR of 12.50%, "TM + NEO-DLN IR + ICI" treatment group performed equally poor as the "TM + C-DLN IR + ICI" in response to the combined treatment.

In conclusion, in contrast to the adjuvant DLN IR, irradiating the DLNs in a neoadjuvant setting was insufficient to overcome the detrimental effect of DLN IR on the efficacy of radioimmunotherapy.

## Adjuvant draining lymph node irradiation improves regional lymph node control, mitigates the growth of a distant (non-irradiated) tumor and allows for the induction of long-lasting tumor-specific immunity

To extend our findings on the level of local control in relation to the scheduling of DLN IR, we assessed the effect of different treatment regimens on additional translationally relevant endpoints. We first assessed the importance of DLN IR for regional control. To this end, we compared the abundance of DLN-infiltrating, luciferase-expressing B16F10 tumor cells performing bioluminescence imaging on the DLNs of tumor-bearing mice treated with different treatment regimens (Fig. 6A). All mice received immunotherapy on days −2, 0 and 2, relative to tumor IR on day 0. In line with the observed negative effect of concomitant and neoadjuvant DLN IR on the efficacy of the combined treatment against the tumor, both treatment schedules also abrogated the development of regional control, as illustrated by 5 out of 7 mice being positive for the presence of tumor cells in the DLNs following the "TM + C-DLN IR + ICI" and "TM + NEO-DLN IR + ICI" treatment regimens (Fig. 6B). A similar response pattern was present in the DLNs of mice in the "Sham IR + ICI" group (9 out of 11), which was in stark contrast to

the regional control achieved by the tumor-only IR and the adjuvant DLN IR, whereby DLN-infiltrating tumor cells could be detected in only 2 out of 8 mice.

Next, in order to investigate the induction of long-term tumor antigen-specific immunological memory, we performed a tumor rechallenge experiment on complete responders from the cohort of B16F10-Luc tumor-bearing mice used in the experiments depicted in Figs. 3 and 4. We performed the rechallenge using either the same (B16F10-Luc) or unrelated, antigenically different (MC38) tumor cells, thus enabling the differentiation between tumor antigen-specific and unspecific tumor rejection (Fig. 6C and Supplementary Fig. 2A). The majority of mice treated with tumor-only IR and delayed DLN IR successfully rejected B16F10-Luc tumors, with a take rate of 25%, 0% and 25% for "TM IR + ICI", "TM + A2-DLN IR + ICI" and "TM + A7-DLN IR + ICI" groups, respectively (Fig. 6C, left). In contrast, all mice rechallenged with unrelated, antigenically different MC38 cells developed tumors (Fig. 6C, right). The baseline take rate of 100% was established in tumor-naïve healthy mice which were injected at the same time as the cured mice (Fig. 6C, gray columns).

To probe the generalizability of our findings and to investigate the potential benefit of delayed DLN IR also for the distant disease control, we compared concomitant and adjuvant treatment regimens using an additional unilateral tumor model (MC38 murine colon carcinoma) and two bilateral tumor models (B16F10 wild type murine melanoma and MC38 murine colon carcinoma), a clinically relevant fractionation schedule (8 Gy ×3) and an alternative widely used ICI (α-PD-1) (Fig. 6D–H, Supplementary Fig. 2B and Supplementary Fig. 3A–O). First, we compared concomitant and delayed DLN IR in mice bearing bilateral subcutaneous tumors derived from murine wild type B16F10 melanoma cells. The previously used luciferase-expressing B16F10-derived tumors are considered highly immunogenic due to the presence of the xenogenic reporter. Therefore, we aimed to investigate whether the beneficial effect of adjuvant DLN IR was present also in the related, but immunologically cold B16F10 wild type tumors. Furthermore, to increase translational significance, we applied a clinically relevant immunomodulatory hypofractionation regimen to the primary tumor and the respective DLNs (8 Gy ×3)[35] and a combined checkpoint blockade approach (α-CTLA-4 + α-PD-1), which is commonly used in advanced malignant melanoma[36] (Fig. 6D). In line with data from the B16 F10-Luc model, delayed DLN IR resulted in a significantly stronger treatment response compared to concomitant DLN IR (Fig. 6E–H and Supplementary Fig. 2B). In the Kaplan–Meier analysis, adjuvant DLN IR increased the median time to reach a cumulative tumor volume of 1000 mm³ by 8 days (Fig. 6F), owing largely to a tumor growth delay in the distant, non-irradiated tumor ("abscopal" effect)[6,7]. However, the adjuvant "TM + A7-DLN IR + ICI" treatment regimen also improved local control in this tumor model, as demonstrated by 12/13 mice achieving a 50% or more decrease in the primary

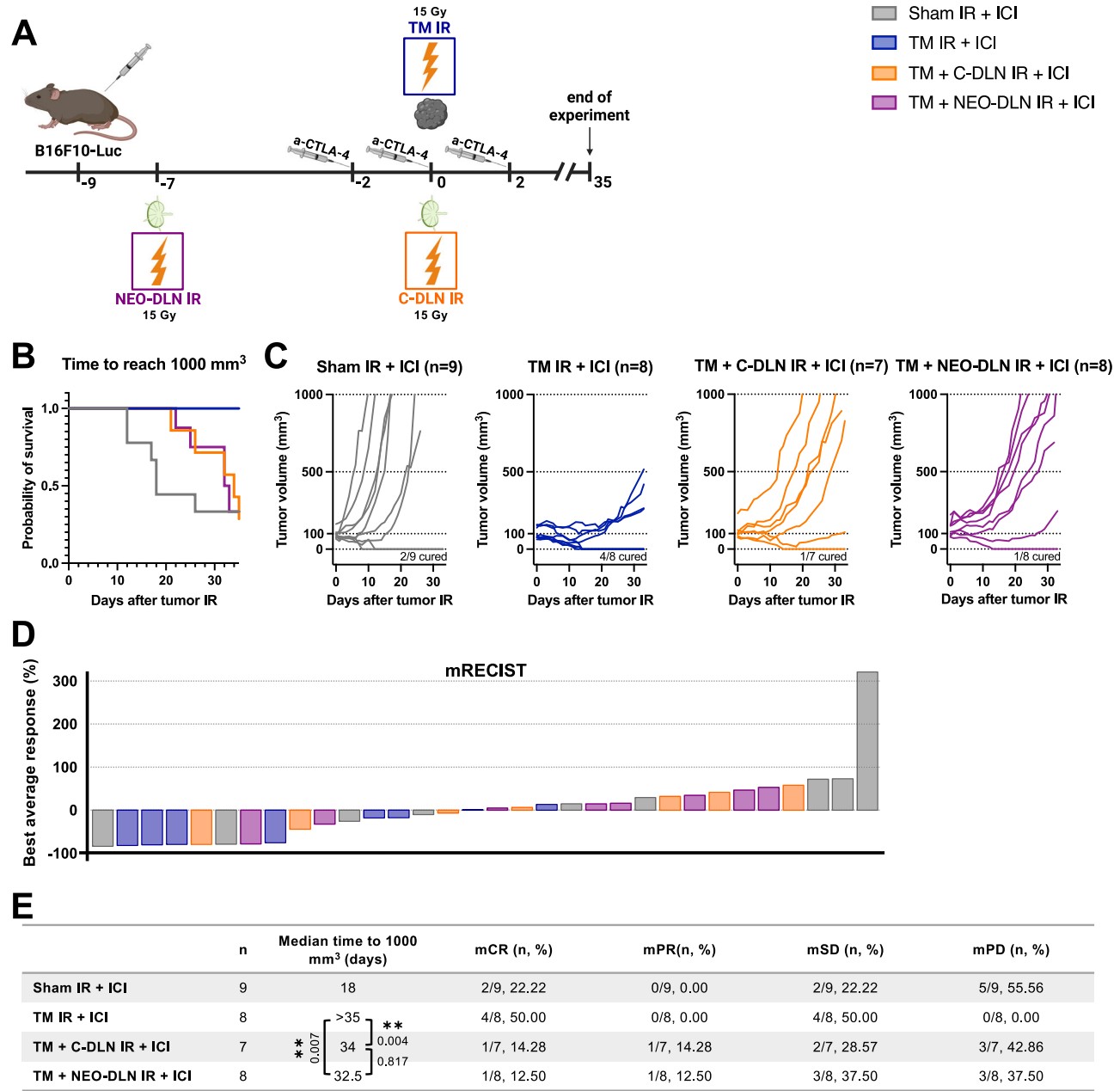

**Fig. 5 | Neoadjuvant draining lymph node irradiation fails to preserve the efficacy of radioimmunotherapy. A** B16F10-Luc tumor-bearing mice received α-CTLA-4 and tumor IR, with or without DLN IR at different timepoints. All groups received α-CTLA-4. "Sham IR + ICI" group (gray) received was sham-irradiated "TM IR + ICI" group (blue) received tumor IR, "TM + C-DLN IR + ICI" group (orange) received DLN IR concomitantly to the tumor IR and "TM + NEO-DLN IR + ICI" (purple) received DLN IR 7 days prior to tumor IR. Tumor growth was followed over 35 days. Treatment response represented by the Kaplan–Meier survival analysis (**B** and **E**), individual tumor growth curves (**C**) and a waterfall plot derived from the mRECIST analysis (**D**). Time to reach 1000 mm³ was used as the endpoint for

Kaplan–Meier analysis. Each line in **C** and each bar in **D** represents an individual mouse. Parameters derived from the mRECIST analysis in **D** and **E** are described in the Methods section. mCR complete response, mPR partial response, mSD stable disease, mPD progressive disease. Number of mice in each group is indicated in the corresponding graph title in **C**. Logrank test (Mantel–Cox) was used to compare the survival curves; corresponding *p* values are displayed in **E**. All *p* values are displayed, with *, ** and *** indicating *p* < 0.05, *p* < 0.01 and *p* < 0.001, respectively. Source data are provided as a Source Data file. Figure **A**, created with BioRender.com, released under a Creative Commons Attribution-NonCommercial-NoDerivs 4.0 International license.

tumor volume following IR (Fig. 6G, green bar), compared to only 4/12 mice treated with concomitant DLN IR (Fig. 6G, orange bar). The comparison of mean tumor volumes confirmed the significant beneficial effect of delayed DLN IR for both the primary (Supplementary Fig. 2B, left) and the secondary tumor (Supplementary Fig. 2B, right).

Next, we assessed the treatment response to concomitant versus delayed DLN IR in mice bearing tumors derived from murine MC38 colon carcinoma cells (Supplementary Fig. 3A–O). In the single tumor model, using α-CTLA-4 and a single high dose of IR (Supplementary

Fig. 3A), both concomitant and delayed DLN IR resulted in a high cure rate, with a trend towards an improved treatment efficacy in response to regimen with the 7-day delay of DLN IR (Supplementary Fig. 3B–E). We hypothesized that in this model, the contribution of delayed DLN IR and thereby enhanced antitumor immunity in response to the combined radioimmunotherapy may become more obvious on the systemic level. Therefore, mice bearing two subcutaneous tumors (Supplementary Fig. 3F) were treated with IR to only one of the tumors (including concomitant or delayed DLN IR) in order to probe the

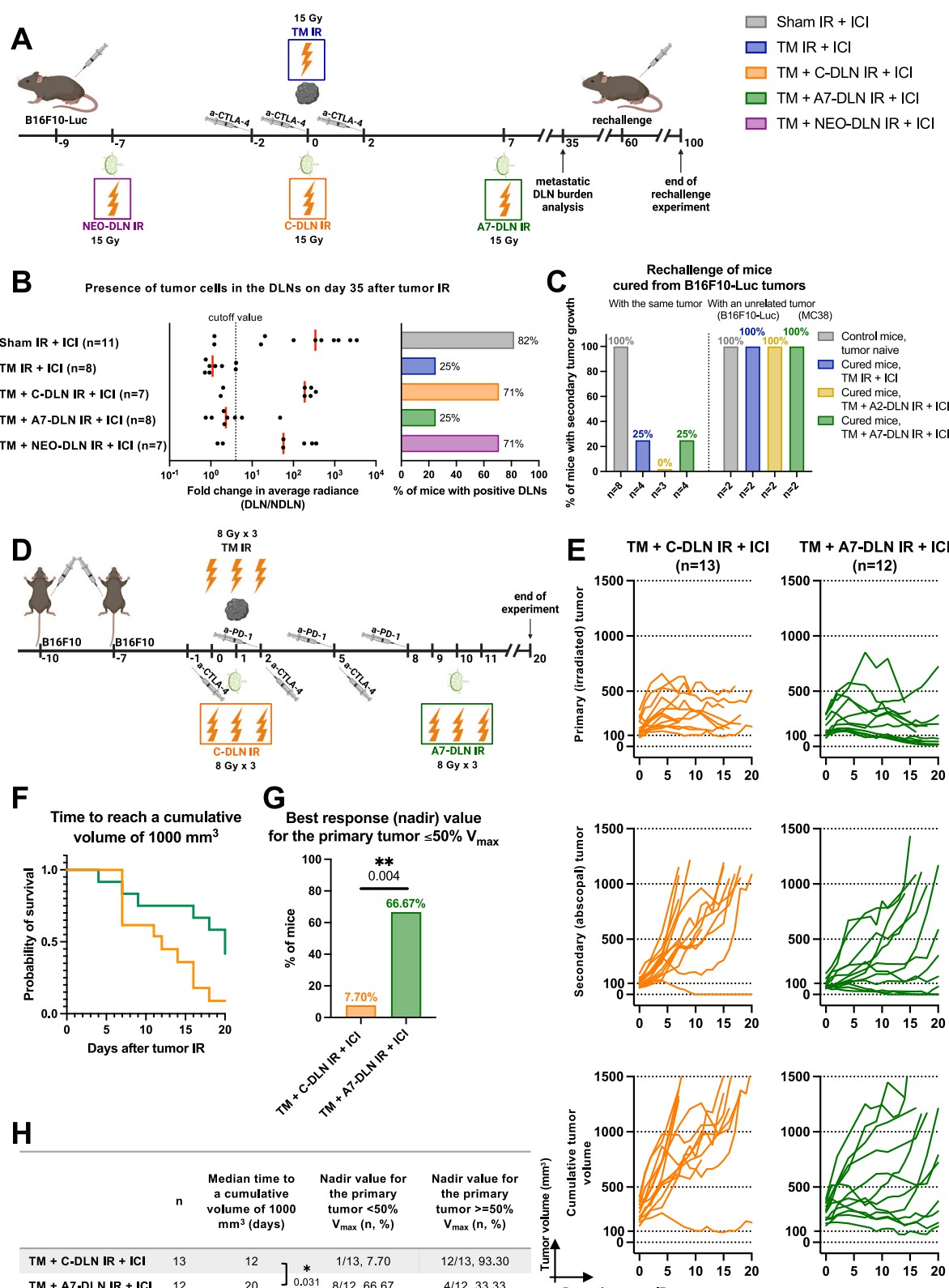

efficacy of the combined radioimmunotherapy regimen also against the distant, non-irradiated tumor, again using α-CTLA-4 and a single high dose of IR. No significant differences were observed in the response of the primary tumor (Supplementary Fig. 3G, top row and Supplementary Fig. 3H, top row). However, a trend towards an improved abscopal treatment response of the secondary, non-irradiated tumor was present (Supplementary Fig. 3G, bottom row and Supplementary Fig. 3H, middle row). The significance of the

observed trend was confirmed in the cumulative Kaplan–Meier survival analysis (time to reach a cumulative tumor volume of 1000 mm³) (Supplementary Fig. 3I, J). These findings were additionally supported by the similar response pattern following a combination of α-PD-1 instead of α-CTLA-4 and a hypofractionated radiotherapy regimen (8 Gy ×3) instead of 15 Gy ×1 in the same bilateral MC38 tumor model (Supplementary Fig. 3K–O). In response to this treatment regimen, the significance of the beneficial effect of delayed DLN IR on the abscopal

**Fig. 6 | Adjuvant draining lymph node irradiation allows for the development of regional control and the induction of long-lasting tumor-specific immunity. A** B16F10-Luc tumor-bearing mice received α-CTLA-4 and tumor IR, with or without DLN IR at different timepoints, as illustrated. For the evaluation of the presence of tumor cells, DLNs were excised on day 35 after tumor IR. For the rechallenge, cured mice were injected on the contralateral flank, using either the same B16F10-Luc or antigenically unrelated MC38 cells on day 60 after tumor IR. Tumor growth was followed for 40 days after the rechallenge. **B** Quantification of tumor cell positivity in the DLNs. Left: Fold change in the average radiance of the DLN compared to the contralateral NDLN. Dotted line indicates the cutoff value for tumor cell positivity, defined as a 300% increase in the signal over the NDLN (fold change >4). Each dot represents an individual mouse. Red line represents the median. Right: Quantification of mice with DLN metastasis, using the cutoff value of 4. Number of mice in each group is indicated in the graph. **C** Percentage of mice with second tumor growth after rechallenging, using either the same B16F10-Luc or antigenically unrelated MC38 cells. Number of mice in each group is indicated in the graph. **D** Mice bearing two B16F10-Luc tumors received α-CTLA-4, α-PD-1 and tumor IR, as

indicated. DLN IR was performed either concomitantly (C-DLN IR, orange), or 7 days after tumor IR (A7-DLN IR, green). All targets received 8 Gy per fraction in three fractions. Tumor growth was followed over 20 days. Treatment response represented by individual tumor growth curves (**E**), Kaplan–Meier survival analysis (**F** and **H**), and the percentage of mice achieving a 50% or more decrease in the primary (irradiated) tumor volume (i.e. nadir value ≤ 50% $V_{max}$) (**G** and **H**). Time to reach a cumulative volume (i.e. the sum of the primary and secondary tumor volume on a given day) of 1000 mm$^3$ was used as the endpoint for Kaplan–Meier analysis. Each line in **E** represents an individual mouse. Number of mice in each group is indicated in **E**. Logrank test (Mantel–Cox) was used to compare the survival curves; corresponding $p$ values are displayed in **H**. Two-sided Fisher's exact test was used to compare the categorical data in **G**. All $p$ values are displayed, with *, ** and *** indicating $p < 0.05$, $p < 0.01$ and $p < 0.001$, respectively. Source data are provided as a Source Data file. Figures **A** and **D**, created with BioRender.com, released under a Creative Commons Attribution-NonCommercial-NoDerivs 4.0 International license.

treatment response of the secondary, non-irradiated tumor was demonstrated in the comparison between the mean tumor volumes (Supplementary Fig. 3L, bottom) and in the cumulative Kaplan–Meier survival analysis (Supplementary Fig. 3N).

Collectively, these findings demonstrate that, compared to concomitant and neoadjuvant DLN IR, delayed DLN IR improves regional and distant disease control and allows for the induction of long-term tumor-specific immunological memory.

## Concomitant and neoadjuvant draining lymph node irradiation induce prolonged lymphopenia in the irradiated lymph node

To identify IR-induced processes in the lymph node contributing to the loss of efficacy of combined radioimmunotherapy, the immune cell compositions of the DLNs following the different treatment regimens were analyzed on days -3, 0 and 4, relative to tumor IR (which corresponds to days 4, 7 and 11 after neoadjuvant DLN IR, respectively) (Fig. 7A and Supplementary Fig. 4A). As in Fig. 4G, the analysis was performed prior to the day of delayed DLN IR in the adjuvant treatment group (day 7 after tumor IR). Thus, the immune cell composition of the DLNs from the "TM + A7-DLN IR + ICI" group corresponds to the one of the "TM IR + ICI" group.

The analysis of the immune cell counts in the DLNs revealed significant immune cell depletion in the lymph nodes irradiated in the neoadjuvant setting, which was apparent already on day 4 after neoadjuvant DLN IR (corresponding to 3 days prior to tumor IR) and did not return to baseline up until day 11 after neoadjuvant DLN IR (corresponding to day 4 after tumor IR) (purple bars in Fig. 7B, C). Hypocellularity was observed in all major T cell subpopulations, with an absolute cell count normalized to the control of $56.78 \pm 12.67\%$, $55.17 \pm 12.59\%$ and $70.68 \pm 15.95\%$ for CD8$^+$, helper (CD4$^+$ FOXP3$^-$) and regulatory (CD4$^+$ FOXP3$^+$) T cells, respectively, on day 4 after tumor IR (Fig. 7C). The slightly more pronounced decrease in the absolute cell counts of CD8$^+$ and helper T cells compared to regulatory T cells led to a decreased proportion of CD8$^+$ and helper T cells within the CD45$^+$ compartment (Fig. 7D). In line with these findings, a decrease in the CD8$^+$ to regulatory T cell ratio was also apparent on day 4 after tumor IR (Supplementary Fig. 4B).

The analysis of the DLNs irradiated concomitantly revealed a similar pattern to that of the DLNs irradiated in the neoadjuvant setting, with immune cell depletion apparent on day 4 after tumor and concomitant DLN IR (orange bars in Fig. 7B, C). Similar to the "TM + NEO-DLN IR + ICI" group, the decrease was not specific to any of the T cell subcompartments, with an absolute cell count normalized to the control of $51.32 \pm 18.45\%$, $48.91 \pm 18.98\%$ and $49.49 \pm 20.38\%$ for CD8$^+$, helper and regulatory T cells, respectively (Fig. 7C). Within the CD45$^+$ compartment, the proportions of the different T cells remained unchanged (Fig. 7D).

In addition to the analysis of immune cell abundance and composition, we probed the phenotype of helper and CD8$^+$ T cells in the

DLNs on day 4 after tumor IR (Fig. 7E, F). Interestingly, the expression of the effector cytokine IFNγ, one of the key mediators of anti-tumor immunity[37], was significantly increased within the helper T cell compartment of mice treated with tumor-only IR (Fig. 7E). In contrast, a higher abundance of the co-inhibitory receptors PD-1, TIM-3 and CTLA-4 was detected on both CD8$^+$ and helper DLN-infiltrating T cells of mice treated with either concomitant or neoadjuvant DLN IR (Fig. 7E, F). Higher levels of CTLA-4 were also present on regulatory T cells residing in the DLNs of mice treated with concomitant or neoadjuvant DLN IR (Supplementary Fig. 4C). No significant differences were observed in the expression of effector cytokines granzyme B or IFNγ within the CD8$^+$ T cell compartment (Fig. 7F). Taken together, these phenotypical changes point towards a developing anti-tumor immune response only present in the non-irradiated DLNs.

Lymphopenia-induced proliferation is a homeostatic mechanism of the immune system driven by survival factors, which ensures rapid recovery of T cell numbers in response to an acute depletion[38,39]. We hypothesized that sustained lymphopenia in the irradiated DLNs could be due to IR-induced disturbance within this tightly regulated process. A significant increase of the proliferation marker Ki67 was detectable in T cells within the DLNs irradiated in the neoadjuvant setting (purple bars, Fig. 7G and Supplementary Fig. 4D). The percentage of Ki67 positive cells was most markedly increased within the CD8$^+$ T cell compartment (Fig. 7G), with a similar trend observed also within the helper and regulatory T cell compartments (Supplementary Fig. 4D). As evidenced in the "TM + NEO-DLN IR + ICI" treatment group, increased proliferation was present already on day 4 after neoadjuvant DLN IR (which corresponds to day -3 relative to tumor IR) (Fig. 7G), with an almost twofold increase in the proportion of proliferating CD8$^+$ T cells in comparison to non-irradiated lymph nodes ($6.74 \pm 0.54\%$ and $3.91 \pm 0.17\%$, respectively). An increase in Ki67$^+$ cells was still present on day 7 after neoadjuvant DLN IR (Fig. 7G, day 0 relative to tumor IR) and remained significant up to day 11 after DLN IR (Fig. 7G, day 4 after tumor IR). Similarly, a marked increase in the proportion of Ki67$^+$ cells within the different T cells compartments was present in the concomitantly irradiated DLNs (orange bars, Fig. 7G and Supplementary Fig. 4D). With the proportion of proliferating CD8$^+$ T cells on day 4 after tumor IR of $9.44 \pm 1.37\%$ compared to $4.55 \pm 1.12\%$ in the "Sham IR + ICI" group, the magnitude of the increase was comparable to the of the "TM + NEO-DLN IR + ICI" group ($8.13 \pm 2.14\%$).

## Irradiation induces changes in the stromal cell compartment of the lymph node

Sustained lymphopenia despite continuous proliferation in the irradiated DLNs might be due to a perturbed communication between the lymph node and circulating lymphocytes. Stromal cells of the lymph node provide a structural network and are also regulators of an immunologically specialized lymph node microenvironment, uniquely

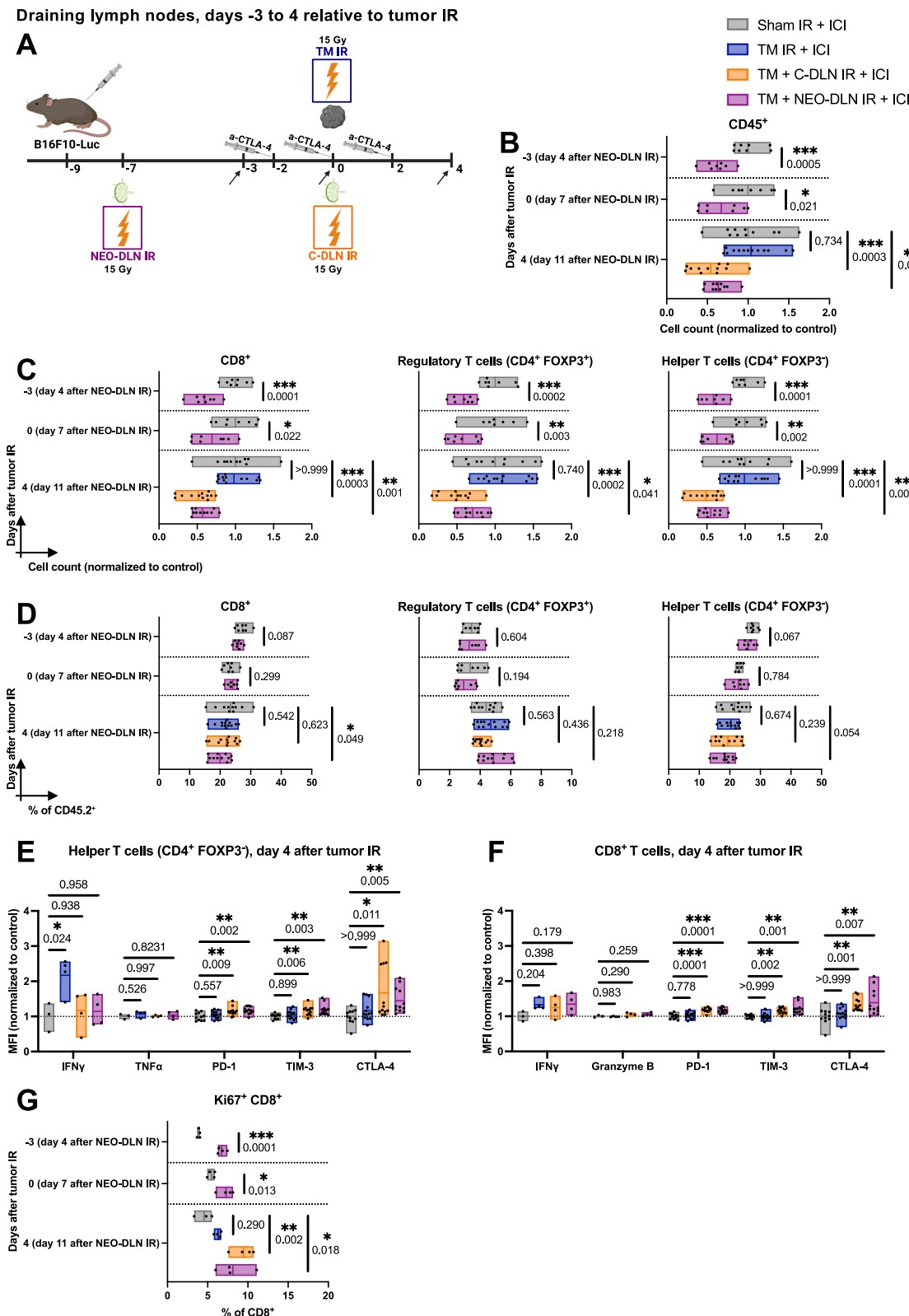

responsible for compartmentalization of antigens, antigen-presenting cells and leukocytes. Thus, an intact stromal cell network is indispensable for the correct immunological function of lymph nodes[40–43]. Given the general lack of data on the effects of IR on the stromal cells of the lymph node, we set forth to investigate how IR affects the composition and the function of distinct lymph node stromal cell subsets, namely fibroblastic reticular cells (FRCs), lymphatic endothelial cells (LECs) and blood endothelial cells (BECs). In order to directly determine the interaction between IR and the stromal cell network, we first performed single high dose lymph node IR in healthy mice (15 Gy) and analyzed the stromal cell subcompartments on day 9 after IR (Fig. 8A). In the immunofluorescence analysis of the immune cells, we observed a disruption in the architecture of B cell follicles (Fig. 8B, top row, white) and mild hypocellularity (Fig. 8B, top row, white and orange). These observations were confirmed in an immunohisto-chemical analysis (Supplementary Fig. 5A–D). Intriguingly, using

**Fig. 7 | Concomitant and neoadjuvant draining lymph node irradiation induce prolonged lymphopenia in the irradiated lymph node. A** All B16F10-Luc tumor-bearing mice received α-CTLA-4. Sham IR + ICI" group (gray) was sham-irradiated, "TM IR + ICI" group (blue) received tumor IR, "TM IR + C-DLN IR + ICI" group (orange) received DLN IR concomitantly to the tumor IR, and "TM + NEO-DLN IR + ICI" group (purple) received DLN IR 7 days prior to tumor IR. Immune cell composition of the DLNs was analyzed at different timepoints (indicated by arrows). Gating strategy is shown in Supplementary Fig. 4A. **B–D** Immune cell composition of the DLN in response to IR. **B** Absolute cell counts of all CD45$^+$ cells. **C, D** CD8$^+$ T cells, regulatory T cells (CD4$^+$ FOXP3$^+$) and helper T cells (CD4$^+$ FOXP3$^-$) represented by cell counts (**C**) and as a percentage of CD45$^+$ cells (**D**). **E, F** Expression of various activation and exhaustion markers on day 4 after tumor IR, expressed as the geometric mean of the fluorescence intensity (MFI), normalized to the average MFI value of the "Sham IR + ICI" group. **E** Helper T cells. **F** CD8$^+$ T cells. **G** Percentage of CD8$^+$ T cells in the DLN positive for Ki67. Each dot represents an individual mouse. Floating bars span from the minimal to the maximal value of each group. Line indicates the mean. $n \geq 8$ for **B–D**, and $n \geq 3$ mice per groups for **E–G** (exact numbers provided in Source Data file). Data were tested for normality using the Shapiro–Wilk test. For data following a normal distribution (all data except as specified below), treatment groups were compared using the two-sided unpaired $t$ test or one-way ANOVA with Holm–Sidak's multiple comparisons test. For non-normally distributed data, comparisons were performed using the two-sided Mann–Whitney test (helper T cells in **C**, day 0; regulatory T cells in **D**, day 0; **G**, day 0) or the Kruskal–Wallis test (CD8$^+$ T cells in C, day 4; TIM-3 and CTLA-4 in **E** and **F**), with Dunn's multiple comparisons test. All $p$ values are displayed, with *, ** and *** indicating $p < 0.05$, $p < 0.01$ and $p < 0.001$, respectively. Source data are provided as a Source Data file. Figure **A**, created with BioRender.com, released under a Creative Commons Attribution-NonCommercial-NoDerivs 4.0 International license.

markers for the different stromal cells subsets, we identified tangible structural changes in the stromal cells of irradiated lymph nodes, particularly within the FRC compartment [represented by podoplanin (PDPN) staining in green, Fig. 8B], which appeared hypointense, with a less interconnected meshwork and a decreased cell surface due to a more rounded morphology. Quantification using flow cytometry revealed a subtle IR-induced decrease of the lymph node stromal cell population (defined as CD45$^-$), mostly attributable to the LECs (CD31$^+$ PDPN$^+$) and partially to the FRCs (CD31$^-$), while the BECs (CD31$^+$ PDPN$^-$) subset remained intact (Fig. 8C and Supplementary Fig. 6A, B). IR also induced changes in several functional markers, most notably an increase in the expression of ICAM-1 on the FRCs and BECs, an increase in the expression of VCAM-1 on FRCs and a trend towards a decrease in the expression of SCA-1 in BECs (Fig. 8D). No changes in the functional markers were observed in the LECs.

A more detailed investigation into the FRCs subsets uncovered a significant decrease in the abundance of medullary reticular cells (MedRCs, defined as PDPN$^+$ CD157$^-$) following IR, while the T zone reticular cells (TRCs, defined as PDPN$^+$ CD157$^+$) remained quantitatively unchanged (Fig. 8E and Supplementary Fig. 6C), suggesting that the observed mild reduction of FRCs in irradiated lymph nodes is primarily attributable to the MedRCs.

Collectively, our investigation of the effect of IR on the stromal cell compartment of the lymph node revealed quantitative and structural changes, prompting the exploration of potential IR-induced functional changes within the stromal cell network as the possible key drivers of the diminished immunological function following DLN IR.

### Lymph node irradiation interferes with the CCR7-CCL19/CCL21 immune cell homing axis

The homeostatic chemokines CCL19 and CCL21, which are constitutively produced by the stromal cells of the lymph nodes guide the migration of circulating, CCR7 receptor-expressing T cells through the high endothelial venules into the lymph nodes[41,44]. Following the observed quantitative and structural changes within the stromal cell network of the irradiated lymph nodes, we hypothesized that lymph node IR could also induce functional changes in the stromal cell subsets crucial for the integrity of the CCR7-CCL19/CCL21 axis. Disrupted axis might subsequently result in the inability of the irradiated lymph node to overcome the IR-induced lymphopenia and to resume its immunological function. Indeed, quantitative analysis of CCL19 on days 2 and 9 after IR (15 Gy ×1, Fig. 9A) revealed a marked reduction of the homeostatic chemokine CCL19 in the irradiated lymph nodes (Fig. 9B, left). On day 2 after IR, CCL19 concentration in the irradiated lymph nodes was 64% lower than in the "Sham IR" group, whereas CCL21 concentration decreased by 47% (Fig. 9B, left and right, respectively). A similar trend was present on day 9 after IR, whereby the concentrations of CCL19 and CCL21 in the irradiated lymph nodes decreased by 40% and 53%, respectively, in comparison to the basal values determined in the sham-irradiated lymph nodes.

A decreased concentration of CCL19 and CCL21 could originate either from a reduction in the number of chemokine-producing cells or from reduced chemokine expression. Despite a slight reduction in the quantity of FRCs following IR, the analysis of FRCs subsets implied that the quantity of the main producers of CCL19 and CCL21, namely TRCs, remained unchanged (Fig. 8C, E). Therefore, we isolated CD31$^-$ FRCs from the irradiated lymph nodes using fluorescence-activated cell sorting and performed quantitative PCR analysis (Fig. 9C). With the value of $0.059 \pm 0.024$ (normalized to *Hprt*) on day 9 after IR, mRNA expression of *Ccl19* in the irradiated lymph nodes was significantly reduced compared to the normalized value of $0.133 \pm 0.065$ measured in the "Sham IR" group, supporting an IR/stress-induced downregulation of chemokine expression of functional significance. Immunofluorescent detection of CCL19 corroborated reduced expression of CCL19 within the FRC network of irradiated lymph nodes (Fig. 9D).

CCR7 is the corresponding receptor for CCL19 and CCL19 and thus an important component of the homing axis. We therefore investigated the expression of CCR7 on CD8$^+$ T cells in lymph nodes which were either sham-irradiated or irradiated with 15 Gy 2 days prior to the resection (Fig. 9E, F and Supplementary Fig. 6D). CD8$^+$ T cells from both treatment groups demonstrated some CCR7 expression (as compared to the isotype control), although the MFI of CCR7 on the surface decreased significantly upon IR (Fig. 9E, F), as did the proportion of CCR7$^+$ cells within the CD8$^+$ T cell compartment (Supplementary Fig. 6D). We hypothesized that this decrease in the cell surface expression of the receptor was a consequence, rather than the cause of the observed lymphopenia in the irradiated lymph nodes, similar to lymphodepleted lymph nodes in mice carrying the paucity of lymph node T cells (*plt*) mutation, which results in a lack of expression of the homeostatic chemokines CCL19 and CCL21[45,46]. To test this hypothesis, we probed the migratory capacity of CD8$^+$ T cells isolated from sham-irradiated lymph nodes and lymph nodes irradiated with 15 Gy 2 days prior to resection, using CCL19 as the chemoattractant in the bottom chamber of a transwell migration setup (Fig. 9G). As predicted, 100 ng/mL of the attractant chemokine CCL19 induced lymphocyte migration in both treatment groups. The number of lymphocytes undergoing migration within the 3 hours of incubation decreased with the decreasing concentrations of CCL19. Notably, already a 50% reduction in the concentration of CCL19 (which corresponds to the relative reduction observed in the irradiated lymph nodes in Fig. 9B) resulted in a significant reduction of the number of migrated cells. Overall, there were no significant differences between the lymphocytes isolated from the irradiated and sham-irradiated lymph nodes.

Taken together, these findings confirm the presence of IR-induced functional changes in the stromal cell compartment of the irradiated lymph nodes, resulting in a decrease of homeostatic chemokines CCL19 and CCL21, which play a crucial role in the immune cell homing to and trafficking through the lymph node.

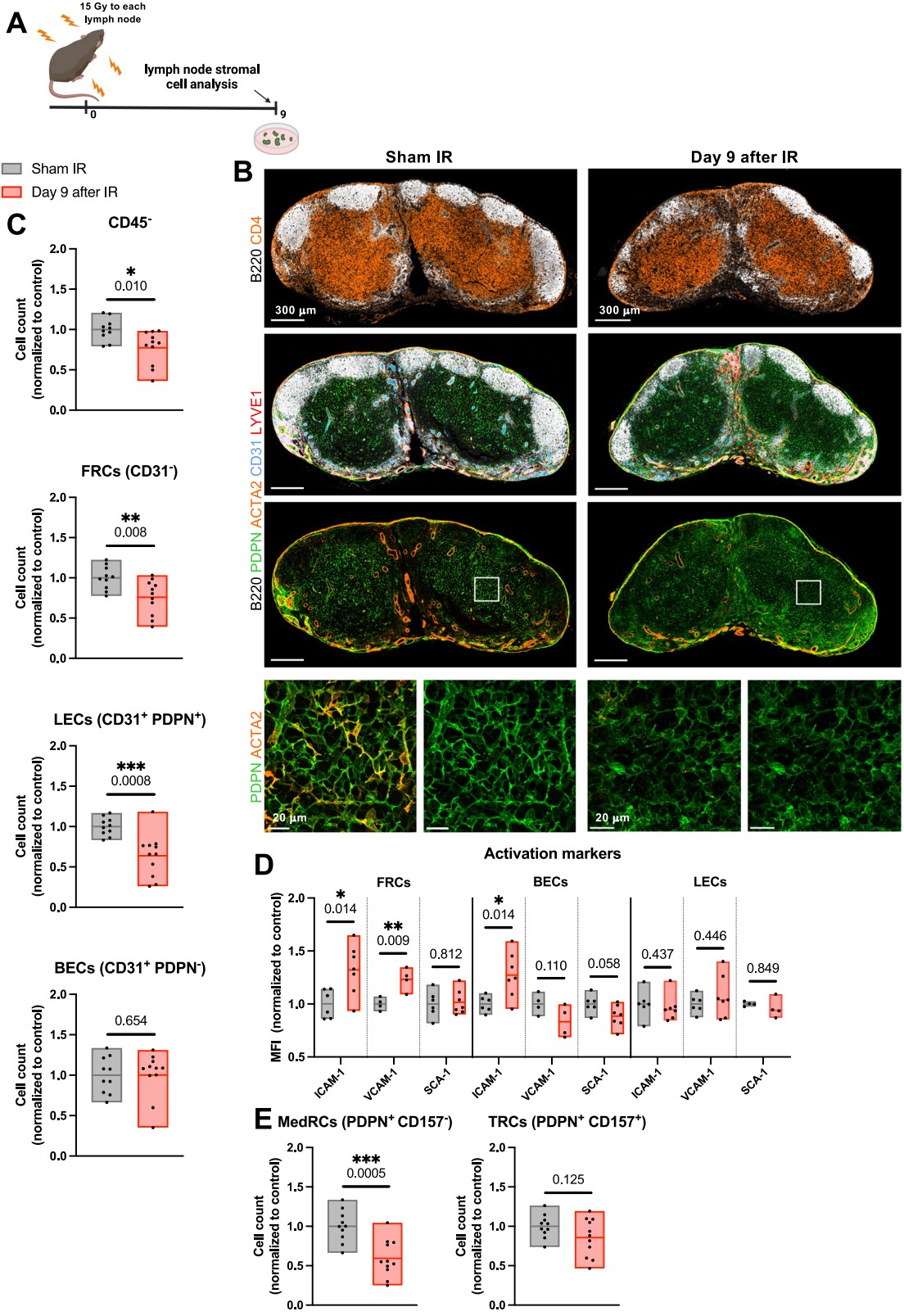

**Draining lymph node irradiation disrupts the CCR7-CCL19/CCL21 axis, which correlates with a reduction in the lymph node-infiltrating cross-presenting conventional type 1 dendritic cells**

Our investigation of the effects of IR on the stromal cell compartment of the lymph nodes in healthy mice revealed an IR-induced interference with the production of CCL19 and CCL21 homing chemokines. We hypothesized that the resulting disruption in the CCR7-CCL19/21

axis could underpin the observed inability of the irradiated DLNs to repopulate and resume their immunological function. As a first step, we set forth to verify that the observed reduction in CCL19 and CCL21 in healthy mice was also present in B16F10-Luc tumor-bearing mice treated with the treatment regimens used throughout this study ("TM IR + ICI", "TM + C-DLN IR + ICI" and "TM + NEO-DLN IR + ICI") (Fig. 10A). Indeed, quantitative analysis of the DLN lysate on day 2 after tumor IR

**Fig. 8 | Irradiation induces changes in the stromal cell compartment of the lymph node. A** Brachial, axillary, and inguinal lymph nodes on both sides of healthy mice were irradiated with 15 Gy. Lymph nodes were harvested on day 9 after IR. **B** Fluorescence microscopy of a section of a sham-irradiated (left column) and an irradiated inguinal lymph node (right column). Sections are stained for B220 (white), CD4 (orange, top row), podoplanin (PDPN) (green), actin alpha 2 (ACTA2) (orange, bottom three rows), CD31 (blue) and LYVE1 (red). Bottom row shows enlargement of areas outlined with white squares. Representative sections from $n = 2$ mice per treatment group, 2 lymph nodes per mouse. **C–E** Flow cytometry analysis of the stromal cell compartment of the lymph node. Gating strategy is shown in Supplementary Fig. 6A. Each dot represents an individual mouse. Floating bars span from the minimal to the maximal value of each group. Line indicates the mean. **C** Top to bottom: Cell counts of all CD45⁻ cells, fibroblastic reticular cells (FRCs), lymphatic endothelial cells (LECs) and blood endothelial cells (BECs). **D** The expression of activation markers ICAM-1, VCAM-1 and SCA-1 on major stromal cell

subsets, expressed as the geometric mean of the fluorescence intensity (MFI), normalized to the average MFI value of the "Sham IR" group. **E** Cell counts of FRCs subtypes, medullary reticular cells (MedRCs) and T zone reticular cells (TRCs). For the sham-irradiated group (gray), $n = 10$ for **C** and **E**, and $n \geq 4$ mice for **D** (exact numbers provided in Source Data file). For the group which received lymph node IR 9 days prior to analysis (red), $n = 11$ for **C** and **E**, and $n \geq 4$ for **D** (exact numbers provided in Source Data file). Data were tested for normality using the Shapiro–Wilk test. For data following a normal distribution, treatment groups were compared using the two-sided unpaired $t$ test (**C**, except for BECs; **D** and **E**). For non-normally distributed data, the comparison was performed using the two-sided Mann–Whitney test (**C**, BECs). All $p$ values are displayed, with *, ** and *** indicating $p < 0.05$, $p < 0.01$ and $p < 0.001$, respectively. Source data are provided as a Source Data file. Figure **A**, created with BioRender.com, released under a Creative Commons Attribution-NonCommercial-NoDerivs 4.0 International license.

(which corresponds to day 9 after neoadjuvant DLN IR) revealed a marked reduction of CCL19 in the DLNs irradiated in both the concomitant and neoadjuvant settings (Fig. 10B, left). A trend towards a decrease in the irradiated DLNs was also observed in the concentration of CCL21 (Fig. 10B, right), suggesting that sustained lymphopenia associated with DLN IR might indeed be due to a perturbation in the CCR7-CCL19/CCL21 axis.

The CCR7-CCL19/CCL21 axis also orchestrates the migration of antigen-carrying DCs from the tumor to the DLNs[41,43,44]. Therefore, we performed detailed immunophenotyping of the DC compartment in the DLNs in response to the different treatment schemes (Fig. 10A, C, D and Supplementary Fig. 7)[47].

In response to DLN IR, both in the neoadjuvant and concomitant setting, we detected a pronounced reduction in conventional type 1 DCs (cDC1s) (Fig. 10C, D). As evidenced in the "TM + NEO-DLN IR + ICI" group, DLN IR led to a rapid depletion of cDC1s, with a 30% and 60% drop in the cell count relative to the control "Sham IR" DLNs on days 4 and 7 after neoadjuvant DLN IR (corresponding to day -3 and day 0 relative to tumor IR), respectively (Fig. 10C, left). This remarkable decrease was also apparent within the cDC compartment, with the proportion of cDC1s dropping from $22.15 \pm 3.61\%$ in the "Sham IR" group to $14.03 \pm 3.18\%$ in the "TM + NEO-DLN IR + ICI" on day 4 after neoadjuvant DLN IR (corresponding to day -3 relative to tumor IR) and from $20.72 \pm 7.36\%$ in the "Sham IR" group to $10.21 \pm 1.96\%$ in the "TM + NEO-DLN IR + ICI" on day 7 after neoadjuvant DLN IR (corresponding to day 0 relative to tumor IR) (Fig. 10C, right). cDC1 depletion remained significant on day 4 after tumor IR (corresponding to day 11 after neoadjuvant DLN IR), when it was also apparent in the concomitantly irradiated DLNs: absolute cell counts normalized to the control were $112.80 \pm 43.29\%$, $34.90 \pm 15.44\%$ and $56.85 \pm 11.24\%$ in the "TM IR + ICI", "TM + C-DLN IR + ICI" and "TM + NEO-DLN IR + ICI" groups, respectively (Fig. 10C, D).

Collectively, these findings strongly suggest that the detrimental effect of concomitant and neoadjuvant DLN IR on the efficacy of radioimmunotherapy might be due to the IR-induced disruption of the CCR7-CCL19/CCL21 homing axis, which results in a substantial reduction of cross-presenting cDC1 in the irradiated DLNs and a subsequent abrogation of T cell priming.

## Discussion

Combining radiotherapy with immunotherapy is a recently developed strategy in cancer treatment based on the propensity of radiotherapy to act as an in situ cancer vaccine[5,9]. Despite encouraging results and strong mechanistic evidence of synergism on the preclinical level, the majority of clinical trials failed to demonstrate the positive effect of the combined treatment. The crucial role of lymph nodes in the development of anti-tumor immunity[19–23] prompted the hypothesis that routine co-irradiation of tumor DLNs might be a major limiting factor to

fully exploit the potential from combining radiotherapy and immunotherapy on the clinical level[13–15].

In this study, we investigated in detail an easily translatable approach to overcome the problem of DLN IR by temporally distancing between IR of the tumor and IR of the DLNs. We used a small-animal image-guided radiotherapy platform to accurately irradiate or spare the DLNs in a murine model of metastatic disease with nodal involvement. Using multiple translationally relevant endpoints, three tumor models and different, clinically relevant radioimmunotherapy combinations, we demonstrated that delayed (adjuvant) DLN IR reverses the detrimental effect of concomitant DLN IR on the efficacy of combined radioimmunotherapy, while simultaneously preserving the beneficial effect of DLN IR on metastatic tumor cell killing. Furthermore, we identified IR-induced quantitative, structural, and functional changes within the stromal cell compartment of the lymph node, which correlated with the disruption of the CCR7-CCL19/CCL21 axis upon DLN IR. Our findings implicate that DLN IR prior to or concomitantly with tumor IR disrupts DLN-to-tumor communication, followed by a disturbance in the tumor-to-DLN immune cell trafficking, ultimately resulting in a severe and sustained reduction in both cross-presenting cDC1s and T cells in the irradiated lymph nodes, thus abrogating tumor IR-induced T cell priming.

A reduced treatment response to (radio)immunotherapy was previously demonstrated upon DLN IR or surgical ablation, and delayed surgical DLN removal was suggested as a treatment option[21,23,24,26]. In this regard, our study confirms the crucial role of DLN sparing as part of combined radioimmunotherapy regimens. More importantly, we reveal the mechanistic link between concomitant DLN IR and the failure of radioimmunotherapy. Furthermore, we demonstrate a treatment regimen which exploits the full potential of the combined treatment even in the setting of high lymphatic metastatic burden or a high risk for microscopic involvement, where DLN IR is unavoidable[16–18]. We demonstrate that keeping the DLNs intact only during the critical time of mounting of anti-tumor immunity in response to tumor IR is sufficient to retain the benefit of DLN IR, while allowing for the vaccine-like effect of tumor IR to fully develop. Delayed DLN IR could thereby also be exploited to increase the potency of systemic anti-tumor immunity and consequently the occurrence of the abscopal effect as also demonstrated in our study.

A functional immune system induces compensatory mechanisms to rapidly reverse lymphopenia[38,39,48]. We and others have previously confirmed a fast reversal of radiation-induced lymphopenia in the blood compartment[27] and in the irradiated lymph nodes[25,34] following localized radiotherapy, albeit using lower doses of IR compared to this study. Thus, the inability of irradiated DLNs to fully recover from IR-induced lymphopenia in our study despite evidence of proliferation, lead us to investigate whether a defect in lymphocyte trafficking might be contributing to prolonged lymphopenia in the irradiated DLNs and consequently to the lack of IR-induced T cell priming. To this end, we

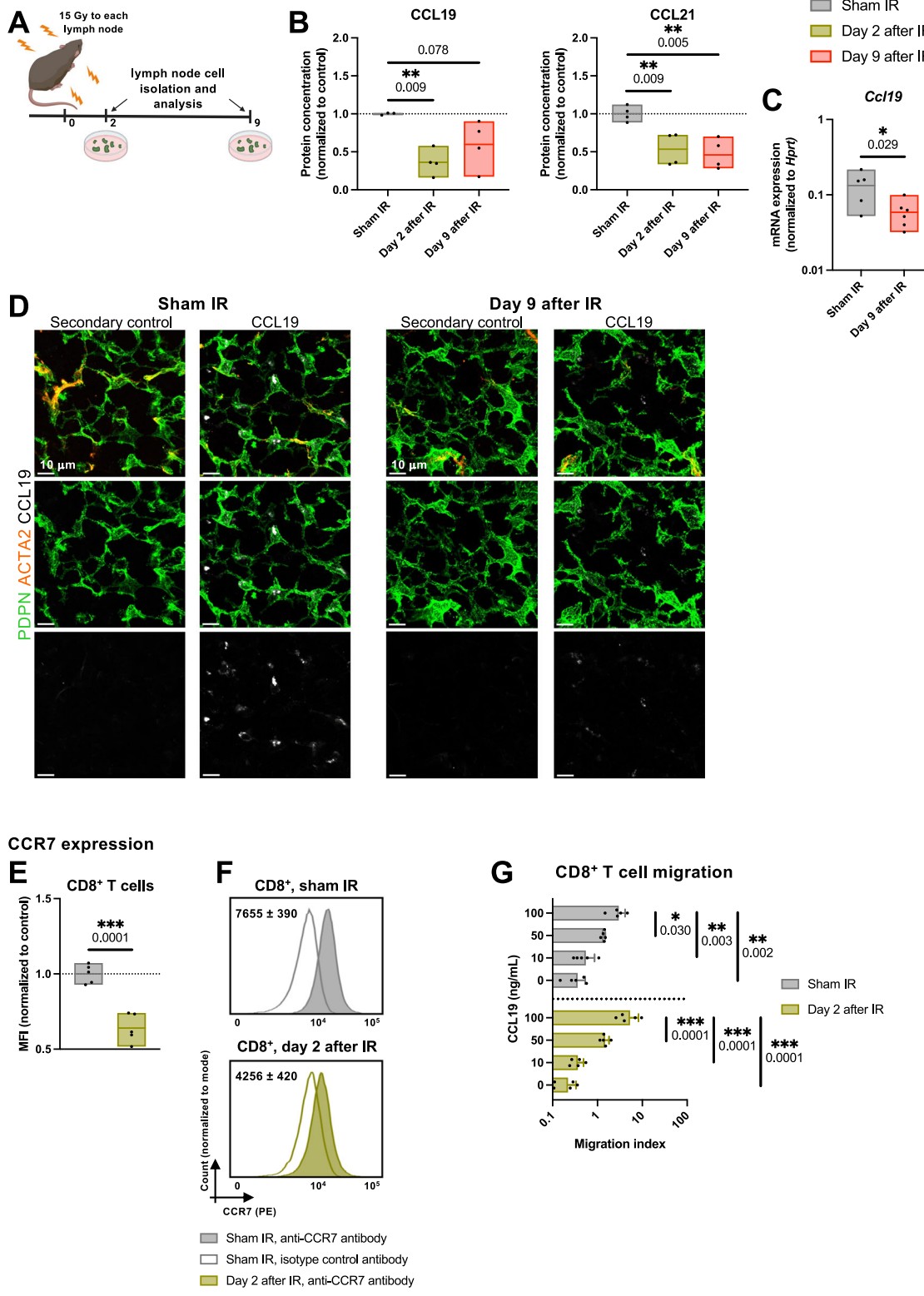

investigated in detail the elusive and previously unexplored interaction of lymph node IR and the stromal cell network, whose structural and functional integrity is a prerequisite for the immunological function of the lymph node[40–43]. To ensure the correct differentiation between direct effects of lymph node IR and potential interference stemming from the presence of a tumor or from tumor IR, we performed a first set of experiments in naïve mice without tumors. Thereby an IR-induced reduction of the homeostatic chemokines

CCL19 and CCL21 in the irradiated lymph nodes was identified. Quantitative PCR analysis revealed that this reduction could largely be attributed to a decreased expression of *Ccl19* in the FRC subset of the stromal cell network of the lymph node and not to a reduced amount of CCL19-expressing cells. Interestingly, we also observed a reduced expression of CCR7 on CD8+ T cells isolated from irradiated lymph nodes. In line with the data on the lymph node composition of CCL19- and CCL21-deficient *plt* mice[45,46], we hypothesized that this decrease in

**Fig. 9 | Lymph node irradiation interferes with the CCR7-CCL19/CCL21 immune cell homing axis. A** Brachial, axillary, and inguinal lymph nodes on both sides of healthy C57BL/6 mice were irradiated using 15 Gy. Lymph nodes were harvested on days 2 (green) and 9 (red) after IR. **B** CCL19 and CCL21 protein concentration in the DLNs, expressed relative to the average value of the sham-irradiated mice (gray). $n = 4$ mice per group. **C** *Ccl19* mRNA expression in fibroblastic reticular cells, normalized to *Hprt*. $n = 6$ mice per group. **D** Fluorescence microscopy of a section of a sham-irradiated (left) and an irradiated inguinal lymph node (right), on day 9 after IR. Sections are stained for podoplanin (PDPN) (green), actin alpha 2 (ACTA2) (orange) and CCL19 (white). Representative sections from $n = 2$ mice per treatment group, 2 lymph nodes per mouse. **E, F** CCR7 expression on CD8$^+$ T cells. **E** Geometric mean of the fluorescence intensity (MFI) of CCR7 on CD8$^+$ T cells, normalized to the corresponding average MFI value of in the sham-irradiated group. **F** Representative histograms. Values in histograms indicate the average MFI of CCR7 on CD8$^+$ T cells (shaded histograms) minus the MFI of the corresponding isotype control (transparent histograms) ± standard deviation (SD). $n = 5$ mice per group. **G** Transwell migration assay of CD8$^+$ T cells isolated from sham-irradiated lymph nodes (gray) and lymph nodes irradiated with 15 Gy 2 days prior to resection (green). Migration index is calculated by dividing the number of migrated cells in the given condition with the basal migration value (i.e. the number of migrated cells towards the bottom chamber containing 10% FBS). $n = 5$ mice per group. Each dot represents an individual mouse. Floating bars in **B, C** and **E** span from the minimal to the maximal value of each group. Line indicates the mean. Bar width in **G** represents the mean value, with error bars indicating the SD. According to the Shapiro–Wilk test, all data followed a normal distribution. Groups were compared using the two-sided unpaired *t* test (**C** and **E**), one-way ANOVA with Holm–Sidak's multiple comparisons test (**B**) and two-way ANOVA with Holm–Sidak's multiple comparisons test (**G**). All *p* values are displayed, with *, ** and *** indicating $p < 0.05$, $p < 0.01$ and $p < 0.001$, respectively. Source data are provided as a Source Data file. Figure **A**, created with BioRender.com, released under a Creative Commons Attribution-NonCommercial-NoDerivs 4.0 International license.

the cell surface expression of the receptor was a consequence, rather than the cause of the observed lymphopenia in the irradiated lymph nodes. In a migration assay, CD8$^+$ T cells isolated from the irradiated lymph nodes followed a physiological response to an increasing concentration of CCL19, suggesting they were functionally capable of responding to the chemokine gradient. Importantly, a 50% reduction in the concentration of CCL19 (which corresponds to the relative reduction we observed in the irradiated lymph nodes) already resulted in a significantly reduced number of migrated lymphocytes, thus supporting the plausibility of the disruption of the CCR7-CCL19/CCL21 axis as the mechanism behind the prolonged lymphopenia in vivo. The occurrence of this effect of IR on the CCR7-CCL19/CCL21 axis was confirmed in our murine model of melanoma.

Importantly, as the only known ligands for CCR7, CCL19 and CCL21 are indispensable for homing of both T cells and DCs to the lymph nodes[40–44,48–50]. Circulating lymphocytes use high endothelial venules (HEVs), a specialized form of blood vessels, to migrate into the lymph node. In contrast, DCs enter the lymph node through the lymphatic route (LECs)[51]. In line with our findings related to the disruption of the CCR7-CCL19/CCL21 axis and a significant reduction in the LECs, irradiated lymph nodes in our model were severely depleted of cross-presenting cDC1, a subset of DCs orchestrating the development of anti-tumor immunity[52]. Thus, immunophenotypical changes observed in the irradiated lymph nodes closely resemble those seen in the CCL19- and CCL21-deficient *plt* mice[45,46,50]. Together with the reduction in the expression of CCL19 and CCL21 in response to IR, these findings strongly indicate that DLN IR disrupts immune cell trafficking into the lymph nodes and consequently interferes with tumor IR-induced T cell priming, which is a crucial mechanism behind combined efficacy of radiotherapy and immunotherapy[4]. In support of this hypothesis, Saddawi-Konefka et al. have recently demonstrated that the tumor response to α-CTLA-4 immunotherapy is dependent on the presence of cDC1 in the tumor-draining lymphatics[23], while Darragh et al. correlated DLN IR to a general decrease in DCs in the DLNs[26]. Therefore, a disruption in cDC1 homing to the DLNs and consequently T cell priming, rather than unspecific IR-induced lymphopenia, might be the dominant mechanism behind the detrimental effect of neoadjuvant and concomitant DLN IR as demonstrated in our study.

In our study, we provide hypothesis-generating data, with immediate implications for the design of upcoming radio-immunotherapy clinical trials. In order to consolidate the translational relevance of our findings, we investigated delayed DLN IR also as part of a clinically relevant immunomodulatory hypofractionation regimen (8 Gy × 3)[35] in combination with the currently most widely used ICI α-PD-1, either alone or as part of a dual checkpoint blockade approach (α-CTLA-4 + α-PD-1), which is an established treatment approach in advanced malignant melanoma[36]. Additional radiotherapy regimens, including other hypofractionated schemes, as well as conventional fractionation, could be investigated in order to increase the generalizability of our data. To fully appreciate the relevance of our mechanistic findings, future studies should include targeted mechanistic investigation to identify the underlying structural and/or functional IR-induced disturbances leading to the IR-induced CCR7-CCL19/CCL21 axis disruption in the treatment response to combined radio-immunotherapy. Investigating in detail the elusive multidirectional interaction between lymph node IR, immune and stromal cells, especially of FRCs as key mediators of T cell homeostasis, might be of particular interest in this context[40–44,48,49]. Several functional marker proteins, e.g. ICAM-1, were upregulated in the irradiated lymph node. Thus, the observed reduction in the production of homing chemokines is not only a general but rather a physiological response to IR-induced stress and/or lymphodepletion, aimed perhaps at allowing for a functional and structural reconstitution of the lymph node prior to resuming its immunological function. On the other hand, although less likely, the reduction could also be a direct effect of IR-induced damage on the integrity of the stromal cells. Such mechanism-oriented studies will be important but challenging, as any kind of manipulations within the CCR7-CCL19/CCL21 axis, such as using knockout mice or neutralizing antibodies, will undoubtedly completely abrogate immune cell trafficking and thus the development of anti-tumor immunity, given the well-established role of the axis in the immune system[40,43,44,49]. Therefore, such and similar experiments cannot be used to clarify the exact mechanistic link between lymph node IR and the ensuing immunological dysfunction.

In summary, our study implies that concomitant DLN IR limits the success of current radioimmunotherapy protocols and proposes a potentially highly beneficial, yet easy-to-implement radioimmunotherapy treatment strategy of delayed DLN IR, which preserves the instrumental function of DLNs at the time of primary tumor IR, while still eradicating the metastatic tumor cells.

## Methods

### Ethics statement

All animal experiments were performed in accordance with the Swiss federal and cantonal laws on animal welfare and approved by the Cantonal Veterinary Office Zurich (ZH113/2020 and ZH141/2023). Maximal permitted tumor burden (defined as a single tumor volume of 1500 mm³ or a cumulative tumor volume of 2000 mm³) was not exceeded in this study.

### Study design

The aim of the study was to investigate whether temporal distancing between IR of the tumor and IR of the DLN maximizes the positive effect of radiotherapy on the anti-tumor immune response, while simultaneously preserving the beneficial effect of metastatic tumor cell killing. The study was performed using individual female tumor-bearing mice who were treated with radiotherapy and/or immunotherapy as the units of study. The study included only female mice in

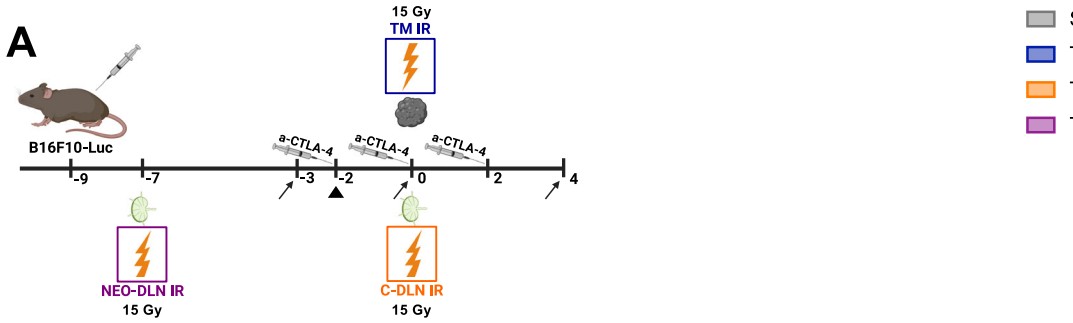

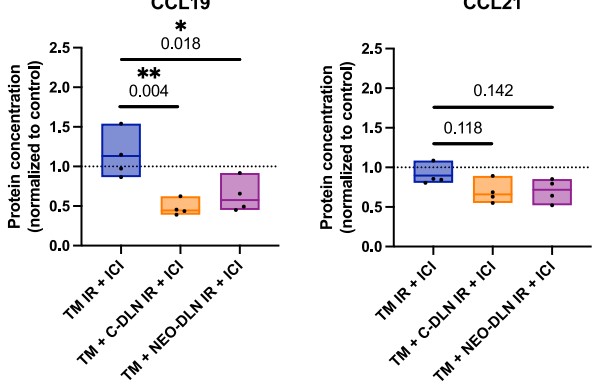

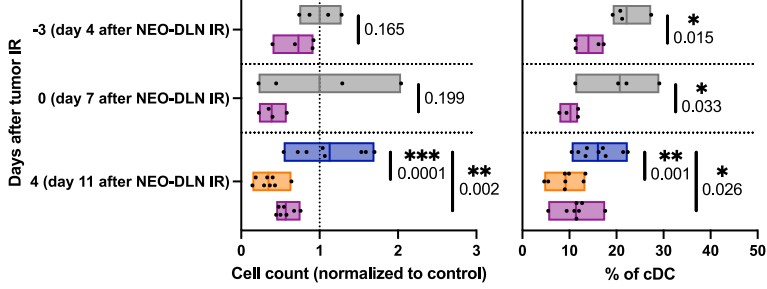

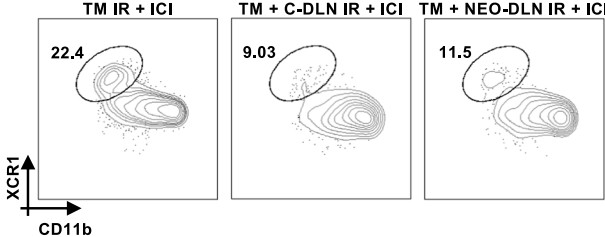

**Fig. 10 | Draining lymph node irradiation disrupts the CCR7-CCL19/CCL21 axis, which correlates with a reduction in the lymph node-infiltrating cross-presenting conventional type 1 dendritic cells. A** All B16F10-Luc tumor-bearing mice received α-CTLA-4. "TM IR + ICI" group (blue) received tumor IR, "TM + C-DLN IR + ICI" group (orange) received DLN IR concomitantly to the tumor IR, and "TM + NEO-DLN IR + ICI" group (purple) received DLN IR 7 days prior to tumor IR. Dendritic cells were analyzed at different timepoints (indicated by arrows). Gating strategy is shown in Supplementary Fig. 7. CCL19 and CCL21 protein quantification was performed on day 2 after tumor IR (arrowhead). **B** CCL19 and CCL21 protein concentration in the DLNs on day 2 after tumor IR (corresponding to day 9 after neoadjuvant DLN IR), expressed relative to the average value of the sham-irradiated mice. *n* = 4 mice per group. **C, D** cDC1s in the DLN displayed as cell counts (**C**, left)

and as a percentage of all cDCs (**C**, right). **D** Representative plots from DLNs on day 4 after tumor IR. Numbers indicate the percentage of cDC1 within the cDC compartment. *n* ≥ 4 mice per group (exact numbers provided in Source Data file). Each dot represents an individual mouse. Floating bars span from the minimal to the maximal value of each group. Line indicates the mean. Data were tested for normality using the Shapiro−Wilk test. All data followed a normal distribution. Treatment groups were compared using the two-sided unpaired *t* test (**C**, day -3 and day 0) and one-way ANOVA with Holm−Sidak's multiple comparisons test (**B** and **C**, day 4). All *p* values are displayed, with *, ** and *** indicating $p < 0.05$, $p < 0.01$ and $p < 0.001$, respectively. Figure **A**, created with BioRender.com, released under a Creative Commons Attribution-NonCommercial-NoDerivs 4.0 International license.

order to increase the robustness of the radiotherapy treatment plan by minimizing the anatomical differences between individual mice. DLNs of mice were either excluded from radiotherapy treatment or were irradiated at different timepoints relative to the tumor IR (in a neoadjuvant, concomitant or adjuvant setting). Stratified randomization based on the tumor volume on a predetermined day after tumor injection was used for treatment group assignment. During the follow up, all persons interacting with the animals were blinded towards the treatment group. For data analysis, each mouse was randomly assigned a number by a person who was not performing the data analysis. Minimal sample size of 6 mice per group was determined based on pilot studies, which were performed to estimate the effect size (change in the absolute tumor volume on day 30 for efficacy studies and change in the percentage of tumor-infiltrating CD8[+] T cells for mechanistic studies) and the dropout rate due to tumor cell rejection, spontaneous regression of the tumor or development of necrosis before reaching the endpoint. All experiments were terminated at predefined endpoints based on previous studies and Swiss federal and cantonal laws on animal welfare. Mice were excluded due to the following reasons: tumor cell rejection, spontaneous regression of the tumor or development of necrosis within 1 week after tumor IR. Outliers were identified by the Grubbs' test and excluded accordingly. Unless otherwise specified, data are pooled or representative from a minimum of two independent experiments.

## Cell lines
The B16F10-Fluc-Puro murine melanoma cell line was purchased from Imanis Life Sciences (catalog number: CL052). The MC38 murine colorectal cancer cell line and the B16F10 murine melanoma cell line were a kind gift from Lubor Borsig (Dept. Physiology, University of Zurich, Switzerland). Cells were authenticated by the providers and passaged up to 5 times prior to the experiments. Cells were cultured in Dulbecco's modified Eagle medium (DMEM, Gibco) supplemented with 10% fetal bovine serum (FBS, Gibco) and 1% penicillin/streptomycin (Gibco) at 37 °C and 5% $CO_2$.

## Animals
7- to 8-week-old female C57BL/6J mice were purchased from Envigo (C57BL/6OlaHsd) and Janvier (C57BL/6JRj). Mice were kept under specific pathogen-free conditions in the animal facility at the University of Zurich. The facility maintains a 12-h light/dark cycle (lights on: 07:00, lights off: 19:00), an ambient temperature of 21–24 °C and a humidity level of 35–70%. For all interventions, the mice were anesthetized with isoflurane (Attane, Piramal Ltd.) (1 L/min oxygen flow rate with 5% isoflurane for induction and 1.5% for maintenance).

## Tumor models
Mice were shaved, followed by a subcutaneous injection on the right flank of the mouse (at the midaxillary line just below the ribcage). $5 \times 10^4$ B16F10-Fluc-Puro, $5 \times 10^5$ MC38 or $3 \times 10^5$ B16F10 cells were injected in 100 µL of a 1:1 mixture of phosphate-buffered saline (PBS) and Cultrex (RGF BME, R&D Systems). For experiments including a secondary tumor, the subcutaneous injection was repeated on the left flank of the mouse, either 3 days (to study the abscopal effect) or 60 days (for rechallenge experiments) after primary tumor injection. Tumor volumes were determined by caliper measurements based on the formula $V = L*W^2/2$, where V is the tumor volume, L is the length (the largest diameter), and W is the width (the diameter perpendicular to L). Quantitative analysis of the tumor growth is detailed in the "Statistical analysis" section below. Detailed information on treatment scheduling is outlined in the figures.

## Identification of draining lymph nodes
Axillary, brachial, and inguinal lymph nodes were identified as the DLNs in our tumor models using Evans Blue dye (Sigma-Aldrich), as

described previously[27]. 100 µL of 1% Evans Blue dye was injected directly into the tumors of untreated mice. After 60 min, the mice were killed using $CO_2$ asphyxiation and the Evans Blue positive lymph nodes were distinguished by eye.

## Bioluminescence imaging
Bioluminescence imaging was performed using the IVIS Spectrum imaging system (Perkin Elmer). Images were analyzed using the Living Image software v.4.7.1 (Perkin Elmer).

For in vivo imaging of the tumors, mice bearing luciferase-expressing B16F10 tumors were injected i.p. with 150 mg/kg IVISbrite D-luciferin (Perkin Elmer) 10 min prior to imaging. Sequential images were acquired for 30 min, using the "Auto" exposure setting, and the sequence with the highest signal was used for analysis.

For ex vivo imaging of the DLNs, mice bearing luciferase-expressing B16F10 tumors were injected i.p. with 150 mg/kg IVISbrite D-luciferin (Perkin Elmer) 7 min prior to euthanasia by $CO_2$. Both DLNs and NDLNs (contralateral axillary, brachial and inguinal lymph nodes) were harvested approximately 10 min after the i.p. injection and immediately transferred into 24-well plates filled with 500 µL PBS supplemented with 300 µg/mL D-luciferin. Sequential images were acquired for 30 min, using the exposure time of 5 min and large binning, and the sequence with the highest signal was used for analysis. Quantitative analysis of the bioluminescence measurements is detailed in the "Statistical analysis" section below.

## Radiotherapy
Radiotherapy was performed using the small animal image-guided radiation platform X-RAD SmART (Precision X-Ray Inc.) and a dedicated small animal treatment planning software SmART-ATP (SmART Scientific Solutions B.V.). Treatment planning was performed as described previously[27]. In brief, the anesthetized mouse underwent CT imaging, followed by precise target identification and designing of an individual treatment plan as described in the "Results" section. Tumors were irradiated using the anterior-posterior/posterior-anterior approach (AP/PA) i.e. two opposing rectangular $8 \times 12$ mm beams, with the isocenter set into the middle of the tumor. Lymph nodes were irradiated using the PA approach, either with the circular 5 mm beam or a rectangular $8 \times 12$ mm beam, with the isocentar set into the tissue-equivalent bed (Superflab, Eckert & Ziegler) on which the mouse was lying during the procedure. Beam characteristics, imaging parameters and quality assurance procedures are detailed in ref. [28].

## Immunotherapy
Anti-mouse α-CTLA-4 (clone 9D9; BioXCell) and anti-mouse α-PD-1 (clone RMP1-14); BioXCell) antibodies were given intraperitoneally (i.p.) at a concentration of 1 µg/µL in 200 µL PBS per dose. Detailed information on treatment scheduling is outlined in the figures.

## Flow cytometry
For immune cell analysis, tumors, brachial, axillary and inguinal lymph nodes were harvested and kept on ice in the flow cytometry buffer (PBS with 2% FBS and 2 mM EDTA) until processing. Tumors were cut into small pieces using scissors and incubated in an orbital shaker in 2 mL/sample of dissociation buffer [DMEM with 10% FBS, 0.5 mg/mL DNase I (Roche) and 1 mg/mL collagenase D (Roche)] for 45 min at 37 °C and 100 rpm. Lymph nodes were disrupted into small pieces using two 26 G needles and digested in 500 µL/sample of dissociation buffer for 15 min at 37 °C and 100 rpm. Following digestion, all organs were passed through a 70 µm cell strainer using a syringe plunger. Absolute cell counts were obtained from the single cell suspension using the EVE automatic cell counter (NanoEntek). Cells were incubated in the extracellular staining mix for 30 min at 4 °C, followed by fixation and permeabilization using the Foxp3/Transcription factor staining buffer set (eBioscience) according to the manufacturer's

instructions. Intracellular staining was performed overnight. For the IFNγ and TNFα staining, single cell suspensions were stimulated ex vivo for 3.5 h at 37 °C in 200 μL/sample of activation buffer [DMEM with 10% FBS, 100 ng/mL phorbol 12-myristate 13-acetate (PMA, Sigma-Aldrich), 1 μg/mL ionomycin (Sigma-Aldrich) and 5 μg/mL brefeldin A (eBioscience)] prior to the extracellular staining. For the CCR7 staining, cells were first incubated with an Fc-receptor blocking antibody for 10 min at 4 °C, followed by an incubation with the CCR7 antibody at 37 °C. After 30 min, the remaining antibodies targeting extracellular markers were added on top and incubated for additional 20 min at 4 °C.

For stromal cell analysis, brachial, axillary, and inguinal lymph nodes were harvested and kept on ice in Roswell Park Memorial Institute 1640 medium (RPMI 1640, Gibco) supplemented with 2% FBS until processing. Stromal cell isolation was initiated by disrupting the lymph nodes into small pieces using two 26 G needles and transferring the pieces into 2 mL/sample of dissociation buffer [RPMI 1640 with 2% FBS, 20 mM 4-(2-hydroxyethyl)-1-piperazineethanesulfonic acid, pH 7.2 (HEPES, Lonza), 200 μg/mL collagenase P (Roche), 30 μg/mL dispase (Roche) and 10 μg/mL DNase I (Roche)]. Tissues were subsequently incubated for 45 min at 37 °C, with resuspension and collection of the supernatant every 15 min. Stromal cells were enriched by incubating the cell suspensions with magnetic anti-CD45 and anti-TER119 beads (MACS MicroBeads, Miltenyi Biotec) for 20 min at 4 °C, followed by passing the suspensions through MACS LS columns (Miltenyi Biotec). Unbound single cell suspensions were subsequently incubated for 20 min at 4 °C with the viability dye eFluor 780 (Invitrogen) and for 30 min at 4 °C with the respective antibodies.

All antibodies used for flow cytometry are listed in Supplementary Table 1. Data were acquired with the Aurora spectral flow cytometer (Cytek; for immune cell analysis) operated with SpectroFlo v.3.0 (Cytek) and the FACSymphony flow cytometer (BD Biosciences; for stromal cell analysis) operated with FACSDiva v.8.0.1 (BD Biosciences). Data processing was performed using FlowJo software v10.8 (BD Biosciences). Quantitative analysis of the flow cytometry data is detailed in the "Statistical analysis" section below.

### Fluorescence-activated cell sorting and quantitative PCR

CD31⁻ cells from axillary, brachial and inguinal lymph nodes were sorted using the FACSMelody cell sorter (BD Biosciences) operated with FACSChorus v1.3 (BD Biosciences). Sorted cells were collected in 100 μL of RNAprotect cell reagent (Qiagen) to preserve the RNA after sorting. Sorting strategy is outlined in Supplementary Fig. 6A. RNA extraction was performed using the Quick-RNA microprep kit (Zymo Research). Reverse transcription and complementary DNA (cDNA) was prepared using the High-capacity cDNA reverse transcription kit (Applied Biosystems) and quantitative PCR was performed using the KAPA SYBR® FAST qPCR master mix (Kapa Biosystems) using the QuantStudio 5 system v.1.5.1 (Applied Biosystems). Expression levels were determined using the *Ccl19* (no. QT02532173, Qiagen) and *Hprt* (no. QT00166768, Qiagen) primers.

### Multiplex cytokine analysis

Axillary, brachial and inguinal lymph nodes were snap-frozen using liquid nitrogen and stored at −80 °C until processing. Once all samples were collected, proteins were extracted using the ProcartaPlex cell lysis buffer (Invitrogen) supplemented with Halt protease and phosphatase inhibitor cocktail (Thermo Scientific) and the BioMasher Standard disposable microhomogenizer (TaKaRa). Following homogenization, samples were passed ten times through an 18G needle. Total protein concentration was determined using the Pierce BCA protein assay kit (Thermo Scientific) and Infinite M200 plate reader (Tecan) according to the manufacturer's instructions. DLN lysates were diluted 1:1000 and 1:10000 for CCL19 and CCL21, respectively. Each chemokine was quantified using the corresponding Mouse

ProcartaPlex Simplex kit (Invitrogen) according to the manufacturer's instructions. Data was acquired with the Bio-Plex 200 system (Bio-Rad).

### Transwell migration assay

Single cells suspensions were obtained from the axillary, brachial and inguinal lymph nodes following the dissociation protocol for flow cytometry (as described above). T cells were isolated using the murine T cell isolation kit (StemCell) according to the manufacturer's instructions.

Transwell migration assays were performed in 96-well plates, using permeable inserts with polycarbonate membranes with 5 μm pores (Corning). Bottom chambers were filled with 235 μL of RPMI 1640 (Gibco) supplemented with 0.5% FBS and the increasing concentrations of recombinant murine CCL19 (0, 10, 50 and 100 ng/mL; PeproTech). For the determination of baseline migration value, RPMI 1640 (Gibco) supplemented with 10% FBS was used as the chemoattractant in the bottom chamber. For the baseline value used for calculating the migration index in the irradiated group (see also "Statistical analysis" section below), lymphocytes were pooled from all irradiated lymph nodes. For the baseline value used for calculating the migration index in the sham-irradiated group, lymphocytes were pooled from all sham-irradiated lymph nodes. Plates were pre-incubated for 30 min at 37 °C, followed by the addition of $2 \times 10^5$ T cells resuspended in 75 μL of RPMI 1640 (Gibco) supplemented with 0.5% FBS to the upper chamber. T cells were allowed to migrate for 3 h at 37 °C and 5% $CO_2$, after which the inserts were carefully removed. Cells which have migrated to the bottom chamber were resuspended in 150 μL of the flow cytometry buffer and quantified following the protocol for flow cytometry described above.

### Immunofluorescence

Brachial and inguinal DLNs were harvested from healthy mice 9 days after IR (15 Gy delivered to each DLN) and fixed for 1 h at room temperature (CCL19 analysis) or 12–24 h at 4 °C (stromal and immune cell analysis) in freshly prepared 4% paraformaldehyde (Merck Milipore) under agitation. After washing in PBS containing 2% FBS and 0.1% Triton X-100 (Sigma), DLNs were embedded in 4% low-melting agarose (VWR International) and sectioned into 40-μM thick sections using the VT1200 vibratome (Leica). Tissue sections were blocked in PBS containing 10% FBS, 1 mg/ml anti-Fcγ receptor (BD Biosciences) and 0.1% Triton X-100 (Sigma) and stained overnight at 4 °C with the respective antibodies in PBS containing 2% FBS and 0.1% Triton X-100 (Sigma). Unconjugated and biotinylated antibodies were stained with secondary antibodies or streptavidin conjugates. All antibodies used for immunofluorescence are listed in Supplementary Table 2.

Confocal microscopy was performed using the confocal microscope LSM-980 (Carl Zeiss), and images were recorded and processed using ZEN 2010 v14.0.18.20 (Carl Zeiss). Imaris v9.2.1 (Oxford Instruments) was used for image analysis.

### Immunohistochemistry

Axillary, brachial and inguinal DLNs were harvested from healthy mice 30 min after DLN IR (15 Gy delivered to each DLN) for the γH2AX biodosimetry, and from tumor-bearing mice on days 2 and 7 after IR (15 Gy delivered both to the tumor and the DLNs) for the assessment of IR-induced changes. Following 24 h of fixation in a 4% formaldehyde solution, DLNs were transferred into 70% ethanol. After routine paraffin wax embedding, consecutive sections (2-3 μm) were prepared and stained with hematoxylin-eosin for histological assessment and by immunohistochemistry for the assessment of markers of interest, using the horseradish peroxidase method. Briefly, after deparaffination, sections underwent antigen retrieval in Tris/EDTA buffer (pH 9) for 20 min at 98 °C. For the γH2AX biodosimetry, sections consecutive to the HE stained section were subsequently incubated with the rabbit

anti-phospho-histone H2A.X (Cell Signaling Technology; diluted 1:200 in dilution buffer, Agilent Dako) for 60 min at room temperature. This was followed by blocking of endogenous peroxidase (Peroxidase blocking solution, Agilent Dako) for 10 min at room temperature and incubation with the secondary antibodies/detection system (EnVision+ system HRP rabbit; Agilent Dako), all in an autostainer (Agilent Dako). Sections were subsequently counterstained with hematoxylin. For the assessment of IR-induced changes, immunohistochemistry for vascular endothelial cells (CD31), T and B cells (CD3, CD45R), macrophages (Iba1) and apoptotic cells (cleaved caspase 3) was performed on sections consecutive to the HE stained section, using previously described protocols[53] and the following primary antibodies: rabbit anti-human CD31 (Abcam), rabbit anti-mouse CD3 (clone SP7; Spring Bioscience Corp., Ventana Medical Systems), rat-anti-mouse CD45R (clone B220/RA3-6B2; BD Pharmingen), rabbit anti-human Iba1 (WAKO), and rabbit anti-cleaved caspase 3 (clone Asp175, 5A1E; Cell Signaling Technology).

Stained sections were scanned using the Nanozoomer 2.0-HT slide scanner (Hamamatsu) and morphometrically analyzed using QuPath v0.3.2[54].

## Statistical analysis

Statistical analysis was performed using Prism v9.5 (GraphPad) and Python v3.7 (Python Software Foundation). Data were tested for normality using the Shapiro–Wilk test. For data following a normal distribution, treatment groups were compared using the two-sided unpaired $t$ test (for two groups), or one-way or two-way ANOVA (for data with one or two independent variables, respectively) with Holm–Sidak's multiple comparisons test (for more than two groups). For non-normally distributed data, comparisons were performed using nonparametric tests (two-sided Mann–Whitney test for two groups and Kruskal–Wallis test with Dunn's multiple comparisons test for more than two groups). For categorical data, contingency tables were analyzed using the two-sided Fisher's exact test. No technical replicates were used to derive statistics in this study. All statistical analyses have been performed using 3 or more biological replicates. For all experiments, the alpha level was set to 0.05. All $p$ values are displayed, with *, ** and *** indicating $p < 0.05$, $p < 0.01$ and $p < 0.001$, respectively.

For the estimation of the tumor cell presence in the DLNs, bioluminescence measurements were analyzed using average radiance (p/s/cm$^2$/sr). The region of interest (ROI) was adapted individually to each lymph node due to size differences. Background signal (defined on a well without a lymph node, filled with 500 μL PBS supplemented with 300 μg/mL D-luciferin) was subtracted from all ROIs. To calculate the fold change in the average radiance, the DLN with the highest background-corrected signal was divided by the background-corrected signal of the corresponding contralateral NDLN. The threshold for lymph node positivity was defined as a 300% increase in the signal over the NDLN (fold change value > 4). Due to the asymmetrical distribution, the data for the estimation of the metastatic DLN burden are plotted using scatter plots displaying each individual data point and the median value.

For the efficacy studies, treatment groups were compared using the Kaplan–Meier survival analysis, mRECIST analysis, AUC analysis and the comparison of mean tumor volume. In the Kaplan–Meier analysis, the endpoint was defined as the time (in days) for the primary tumor volume or the cumulative tumor volume (defined as the sum of the primary and secondary tumor volumes on a given day) to reach 500 or 1000 mm$^3$, as detailed in the corresponding figure legends. Logrank test (Mantel–Cox) was used to compare the survival curves. The mRECIST methodology was adapted from[55]. The tumor volume change $\Delta V_d = 100 \times \frac{V_d - V_{start}}{V_{start}}$ and the average tumor volume change $\Delta \bar{V}_d = \frac{\sum_0^d \Delta V_d}{d}$ were calculated for each day $d$ from the day of tumor IR

until the end of follow up. Best response (BR) was defined as the minimum value of $\Delta V_d$ for $d \geq 7$. Best average response (BAR) was defined as the minimum value of $\Delta \bar{V}_d$ for $d \geq 7$. Response calls were defined as follows: complete response, mCR: BR < -95% and BAR < −40%; partial response, mPR: BR <−50% and BAR <−20%; stable disease, mSD: BR < 35% and BAR < 30%; progressive disease, mPD: not otherwise categorized. For the AUC analysis, the AUC value was calculated for each mouse by integrating the tumor volume values over the duration of the study. To partially correct for the mice that reached the endpoint prior to the end of the study, the curve was extended horizontally by repeating the last measured value until the end of the study, as proposed by Duan et al.[56]. Due to the asymmetrical distribution, the data for the AUC analysis are plotted using bar graphs displaying each individual data point and the median value. For the comparison of mean tumor volumes, statistical analysis was performed at the latest timepoint at which at least two survivors were present in each experimental group. For mice which have reached the termination criteria prior to this timepoint, the last measured tumor volume was used for the statistical analysis.

For flow cytometry, cytokine analysis and quantitative PCR, the data are plotted using floating bars which extend from the minimal to the maximal individual data point, with the line at the mean value. Absolute cell counts were calculated by multiplying the value "% of Total" for the population of interest with the absolute cell count. For the tumors, cell counts were normalized to the tumor mass. Cell counts for lymph nodes, as well as the mean fluorescence intensities (MFIs) and cytokine concentrations, were normalized to the average value of the control samples for each individual experiment. For the CCR7 staining, MFI values of the corresponding isotype controls were subtracted from the MFI values of the samples.

For the transwell migration assay, "migration index" for each sample was calculated by dividing the number of migrated lymphocytes in the sample with the corresponding baseline migration value (see "Transwell migration assay" section above for the definition of the baseline migration value).

## Reporting summary

Further information on research design is available in the Nature Portfolio Reporting Summary linked to this article.

## Data availability

The authors declare that all relevant data supporting the findings of this study are available within the paper and the Supplementary Information file. Source data are provided with this paper.

## Code availability

The code for the adapted mRECIST methodology is available through the Zenodo repository: https://doi.org/10.5281/zenodo.11526666.

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

## Acknowledgements

We thank Onur Boyman and Panagiotis Balermpas for helpful discussions, Sabina Wunderlin. Giuliani for technical support with immunohistochemistry, and the Cytometry Facility of University of Zurich for technical support with flow cytometry. This work was supported by the Swiss Academy of Medical Sciences national MD-PhD grant (MD-PhD-4820-06-2019 to I.T.), the Swiss Cancer Research Foundation grant (KFS-5301-02-2021 to M.P.) and the Swiss National Science Foundation (310030_189285 to M.P.). Figures 1A, D, 2A, 3A, 4A, 5A, 6A, D, 7A, 8A, 9A and 10A, and Supplementary Fig. 3A, F, K were created with BioRender.com, released under a Creative Commons Attribution-NonCommercial-NoDerivs 4.0 International license.

## Author contributions

Conceptualization: I.T., B.L., M.P.; Methodology: I.T., C.Y., L.K., H.W.C., A.S.F.; Formal analysis: I.T.; Investigation: I.T., C.Y., L.K., I.V., S.R., A.S.F., H.W.C., R.W., A.K.; Resources: M.P., M.G., A.K., B.L.; Visualization: I.T., L.K.; Funding acquisition: M.P., I.T.; Supervision: M.P.; Writing – original draft: I.T., M.P.; Writing – review & editing: I.T., B.L., M.P.

## Competing interests

The authors declare no competing interests.
