## [Peer Review File · Nature Communications]

Delayed tumor-draining lymph node irradiation preserves the efficacy of combined radiotherapy and immune checkpoint blockade in models of metastatic diseaseREVIEWERS' COMMENTS:

Reviewer #1 (Remarks to the Author): with expertise in cancer radiotherapy, immunotherapy

The study by Telarovic, Pruschy and collaborators addresses the possibility of improving the efficacy of combining radiation and immune checkpoint inhibitors (ICI) by the rationale selection of the timing for irradiation, prior to (neoadjuvant) or after ICI administration (delayed, adjuvant), as well as using conventional radiation of the compromised tumor draining lymph nodes (DLN) concomitant with the primary lesions, or using lymph node sparing approaches. They have used their recently published murine model of metastatic disease in which they have recently shown that radiation-induced lymphopenia does not impact treatment efficacy (Neoplasia 2022 Sep;31:100812), and now show that delayed (adjuvant), but not neoadjuvant DLN IR overcomes the detrimental effect of concomitant DLN irradiation combined with ICI. They provide some evidence suggestive of a role for the disruption of the CCR7-CCL19/CCL21 homing axis as a mechanism for the detrimental effect of DLN irradiation. The authors put forward the idea concomitant and neoadjuvant DLN irradiation abrogates regional control, while delayed DLN irradiation allows for the development of regional control and the induction of long-term tumor-specific immunological memory.

The study is of high translational potential, but major weaknesses may preclude its consideration for publication in Nature Communications. These include the limited novelty, relevance of their animal model to human cancer, the selection of ICI, the fact that most of the information comes from a single experimental model system, and the superficial nature of the studies on the potential role of the CCR7-CCL19/CCL21 homing axis as the basis of their observations.

Specifically:

- Several recent studies have reported similar findings, using surgery and radiation, and even radioimmunotherapy clinical trials sparing lymph nodes based on these findings have been already initiated. The authors acknowledge these studies, but they need to address

head on the novelty (or lack thereof) of their current manuscript. In addition, many of the figures are similar to their analysis of radiation-induced lymphopenia, which they have already reported (Neoplasia, 2022).

- The authors may need to explain the relevance of their animal model to human cancer. The entire study involves primarily B16F10-Luc cells. The parental B16F10 are very immune evasive and ICI resistant. One can wonder whether the responses observed may relate to the expression of Luc in these cells? In addition, the tumor growth is extremely rapid, which may prevent mounting the full spectrum of innate and adaptive immune mechanisms prior to succumbing from the primary lesions instead of metastatic disease. The very few studies using MC38 cells lack rationale, and the data are not aligned with their results using B16F10 cells. They can only see an impact of delayed DLN radiation on ICI response when addressing the abscopal effect but not the primary lesions.
- While anti-CTLA-4 is now an accepted treatment modality for melanoma and other cancers, its high dependence on heightening the DC-CD8 T cell synapse makes the results not unexpected. That said, the rationale for the use of anti-CTLA-4 vs anti-PD-1/PDL-1, the most widely used ICIs, should be included.
- The potential role of the CCR7-CCL19/CCL21 homing axis as the basis of their observations is correlative in nature, based on decreased levels of the corresponding ligands after DLN irradiation, rather than mechanistic.

Reviewer #2 (Remarks to the Author): with expertise in cancer radiotherapy, immunotherapy

The manuscript by Telarovic et al deals with the important question of the consequences of elective nodal irradiation in the efficacy of radiation/immunotherapy combinations. One strength of this work is the use of a very precise murine model of image-guided lymph node irradiation in metastatic melanoma. However, the main findings (detrimental effect of concomitant DLN irradiation in immune activation and therefore the efficacy of radio-immunotherapy approaches, that can be avoided by delaying DLN irradiation) have long

been a concern in the field and have recently been formally proven in other preclinical studies, in particular Marciscano et al 2018 and Darragh et al 2022. Similarly, the role of type I DCs was described in a model of lymphatics-preserving immunotherapy treatment (Saddawi-Konefka). All these studies are known to the authors and cited in the references.

Other issues are:

- Local vs regional recurrence vs distant or true metastatic growth are poorly differentiated. For example, despite claiming the use of a metastatic melanoma model, most results are related to growth of the primary tumor. Fig 4H-J introduces a two-tumor (MC38) model and reports survival advantage for the adjuvant DLN IR regimen, but it is unclear whether this is due to primary or abscopal tumor growth delay. Fig 6B shows DLN tumor burden, which is referred to as “metastatic” or “regional” in different sections. This could be an important result, however a single experiment is reported and the adjuvant IR group seems to have less mice than the other groups, therefore statistical significance is unclear.
- Some legends are confusing. Eg in Fig 3 “Sham IR” should be renamed “No IR” if mice received no radiation.
- Authors should focus on what could be novel in this study compared to previous publications. The CCL19 data seem to be novel but limited and of unclear translational value, although could be further developed. One interesting finding is that in contrast to radio-immunotherapy, the response to radiation alone was not affected by concomitant DLN IR. This is interesting because the “in situ vaccine” concept for IR is commonly accepted in the field. Lack of effects on IR efficacy by concomitant DLN IR could suggest a lack of involvement of the immune system for local response to IR, or that local immunity was sufficient for primary tumor responses. Are T cells required for efficacy? In either case the potential consequence would be that delaying DLN irradiation could be required for immune-radiotherapy but not for radiation alone. Authors could test whether this is also the case in other models and/or radiotherapy regimes.

Reviewer #3 (Remarks to the Author): with expertise in cancer radiotherapy, immunotherapy

In the manuscript “Delayed tumor draining lymph node irradiation as a new paradigm for radiotherapy-immunotherapy combinations in metastatic disease” (#NCOMMS-23-36401), Telarovic and colleagues investigated whether irradiation of the tumor draining lymph node (TDLN) in neoadjuvant, concomitant or in adjuvant setting modified the response of radiation (RT) with immune checkpoint blocker (ICB). The rationale behind this study relies on the fact that clinical trial testing RT+ICB failed in clinic to generate systemic anti-tumor immunity presumably due to the irradiation of TDLN.

While in preclinical models, it is doable to spare TDLN, in clinic this practice can be questioned due to the metastatic spread in the lymph nodes. Therefore, in metastatic setting, the irradiation of TDLN might be mandatory. Therefore, authors investigated how irradiation of TDLN at different time-point affect RT-induced anti-tumor immunity in the context of ICB and concluded that delayed (adjuvant) irradiation of the TDLN overcomes the detrimental effects of concomitant irradiation of TDLN to induce T-cell mediated responses.

The message of this article is important, novel and within the scope of Nature Communications. I have NO concerns about image manipulation or data fabrication. However, some of the conclusions are not supported by the dataset, and hence must be appropriately revised (or additional experiments performed in this respect). Alongside, some minor issues emerged that need to be addressed, as detailed here below.

Comments:

- Data obtained with the MC38 model is less convincing that the in vivo data obtained with the B16F10-Luciferase model. It is well known that Luciferase is highly immunogenic (at least in preclinical models of breast cancer; PMID: 38798322). Therefore, it is possible that the difference in immunogenicity between these two models is only due to the addition of Luciferase. This potential confounder is also supported by the fact that rechallenge experiment worked in B16F10Luc mice but not in MC38 ones. Authors must repeat in vivo experiments using the B16F10 wild-type model to ensure that their data are not a consequence of the immunogenicity from Luc.
- Along similar lines, authors rechallenged B16F10-Luc surviving mice with fresh inoculum of

B16F10-Luc but it would have been more convincing to rechallenge with B16F10-wild type.

- Authors mentioned that they rechallenged mice from Fig.3 and Fig.4 at day 60. However, sometimes, tumor regrowth occurs around day 80-90. Can authors justify why they decided to rechallenge at that time point and not wait an additional 30-40 days prior to rechallenge to ensure no tumor regrowth?

- T cells depletion is needed to confirm the role of T cells in their system.

- Fig 1G: it seems that gH2AX staining is increase in the non-tumor draining lymph nodes (NTDLN) in the conditions Tumor IR and Tumor IR+DLN IR. Can authors indicate which (NTDLN) they analyzed and provide semi-quantification of their staining? Also, I could not find how many mice neither how many picture were taken to assess the gH2AX staining. Please clarify.

- Authors mentioned that they irradiated by 15Gy. It is not clear if this is the total dose meaning that the mice received 5Gy in the tumor + 5Gy in the TDLN (3x5Gy) or if the mice received 15Gyx3. Please clarify.

- The radiation dose regimen used is not clinically relevant and is not justify in their manuscript. Authors not only should justify their choice but should consider running an experiment with a clinically relevant dose fractionation.

- Authors have performed flowcytometry to assess the T cells compartment throughout their study. However, the definition of Tregs is incomplete in mice. This cell subtype should be defined as live CD45+ FoxP3+CD25+ which is not what is presented in this manuscript (live CD45+ FoxP3+).

- Authors should provide a more in depth characteristic of their T cell phenotype than just only look at CD8 expression. Meaning: do they express activation markers? How about PD-1 expression?

- The disruption of the CCR7-CCL19/CCL21 with radiation of the TDLN is weak. Only 1 experiment is suggesting this mechanism (Fig. 7). To strengthen their findings, authors should disrupt this axis in the RT+ICB animal (no irradiation of the TDLN) to determine whether the tumor control and recruitment of CD8 T cells in RT+ICB animals is due to the homing of DC1 via CCR7-CCL19/CCL21

Reviewer #4 (Remarks to the Author): with expertise in cancer radiotherapy, immunotherapy

this is a very interesting topic addressed by this study since the optimal scheme of radio immunotherapy in terms of tumor volume and lymph node irradiation is poorly defined. A better understanding of the interplay between tumor lymph node irradiation and tumor response to immunoradiotherapy should contribute to improve the results of such combinations in the clinic.

The manuscript reads well overall and the experiments are adequately described providing evidence that delayed lymph node irradiation is superior to neo adjuvant or concomitant irradiation when anti CTLA-4 is added to radiotherapy in terms of local control and systemic immunity.

Some aspects remains to be clarified :

-what is the evidence of nodal tumor involvement dependency in these models? Is tumor lymph node involvement a prerequisite for the benefit of delayed lymph node irradiation?

-In order to maximize the clinical relevance of these data, is there a way to add an experiment that would be more compatible with the radiotherapy regimens performed into the clinic since it is likely that the single dose radiation schemes will not be applicable to tumors where elective lymph node irradiation is performed (ie fractionated irradiation)

-the identification of the role of CCR7-CCL19/CCL21 axis is very interesting but functional data would strengthen the assumption. Manipulation of chemokines using ko mice and or administration of chemokines or blocking antibodies in the same experiments could contribute to reinforce their statements. the statements in the abstract and the manuscript conclusion should be adapted accordingly.

The interpretation of the experiments in the body of the manuscript could be streamlined in order to ease the reader's understanding as well as the figures legends.

Manuscript by Irma Telarovic, Carmen S.M. Yong, Lisa Kurz, Irene Vetrugno, Sabrina Reichl, Alba Sanchez Fernandez, Hung-Wei Cheng, Rona Winkler, Matthias Guckenberger, Anja Kipar, Burkhard Ludewig, and Martin Pruschy titled “Delayed Tumor Draining Lymph Node Irradiation as a New Paradigm for Radiotherapy-Immunotherapy Combinations in Metastatic Disease” (NCOMMS-23-36401)

We thank the reviewers for their assessment of our manuscript. We appreciate their constructive suggestions. We have now revised our manuscript according to the reviewers' comments.

Please note the following nomenclature used for referring to the figures:

Fig. 1 refers to main Figure 1 of the manuscript.

Fig. S1 refers to Supplementary Figure S1 of the manuscript.

Fig. P1 refers to Figure 1 of these point-by-point responses.

Point-by-point responses**Reviewer #1**

The study by Telarovic, Pruschy and collaborators addresses the possibility of improving the efficacy of combining radiation and immune checkpoint inhibitors (ICI) by the rationale selection of the timing for irradiation, prior to (neoadjuvant) or after ICI administration (delayed, adjuvant), as well as using conventional radiation of the compromised tumor draining lymph nodes (DLN) concomitant with the primary lesions or using lymph node sparing approaches. They have used their recently published murine model of metastatic disease in which they have recently shown that radiation-induced lymphopenia does not impact treatment efficacy (Neoplasia 2022 Sep;31:100812), and now show that delayed (adjuvant), but not neoadjuvant DLN IR overcomes the detrimental effect of concomitant DLN irradiation combined with ICI. They provide some evidence suggestive of a role for the disruption of the CCR7-CCL19/CCL21 homing axis as a mechanism for the detrimental effect of DLN irradiation. The authors put forward the idea concomitant and neoadjuvant DLN irradiation abrogates regional control, while delayed DLN irradiation allows for the development of regional control and the induction of long-term tumor-specific immunological memory.

The study is of high translational potential, but major weaknesses may preclude its consideration for publication in Nature Communications. These include the limited novelty, relevance of their animal model to human cancer, the selection of ICI, the fact that most of the information comes from a single experimental model system, and the superficial nature of the studies on the potential role of the CCR7-CCL19/CCL21 homing axis as the basis of their observations.

Specifically:

- Several recent studies have reported similar findings, using surgery and radiation, and even radioimmunotherapy clinical trials sparing lymph nodes based on these findings have been already initiated. The authors acknowledge these studies, but they need to address head on the novelty (or lack thereof) of their current manuscript. In addition, many of the figures are similar to their analysis of radiation-induced lymphopenia, which they have already reported (Neoplasia, 2022).

We thank the reviewer for their comments. There are several suggestions and issues raised by this reviewer with which we agree, and we have addressed them as part of our extensive revisions (see details below). We thank the reviewer for these helpful insights, which we believe have contributed to the strengthening our findings.

However, we would also like to bring forward several points mentioned by the reviewer with which we strongly disagree. The issue of the resemblance to our previous study (Telarovic et al., Neoplasia, 2022; <https://doi.org/10.1016/j.neo.2022.100812>) is unsubstantiated: our former study addresses a completely different research question and uses a completely

different tumor model and treatment regimen, in contrary to what the reviewer states. The murine model of metastatic disease was developed specifically for the current study and the clinical setting which is addressed as part of it. As such we also disagree with the statement of the reviewer that “many of the figures are similar to their analysis of radiation-induced lymphopenia, which they have already reported (Neoplasia, 2022)”, as stated by the reviewer. Furthermore, we would like to point out that our research addresses a clinical situation which was not considered in the previous studies, namely the situation in which lymph node sparing is **not** possible due to the spread of the disease. We investigate an alternative possible approach in this situation, which allows for the retention of the clinically unavoidable lymph node irradiation, while simultaneously allowing for the development of tumor irradiation-induced anti-tumor immunity.

Furthermore, we added a substantial new set of data and insights on the CCR7-CCL19/CCL21 regulation in response to irradiation in the revised manuscript. We believe that the additional points related to the novelty, the animal model, the selection of the ICI, additional experimental model system and the studies on the role of the CCR7-CCL19/CCL21 homing axis are sufficiently addressed in our revised manuscript, as detailed below.

- The authors may need to explain the relevance of their animal model to human cancer. The entire study involves primarily B16F10-Luc cells. The parental B16F10 are very immune evasive and ICI resistant. One can wonder whether the responses observed may relate to the expression of Luc in these cells? In addition, the tumor growth is extremely rapid, which may prevent mounting the full spectrum of innate and adaptive immune mechanisms prior to succumbing from the primary lesions instead of metastatic disease. The very few studies using MC38 cells lack rationale, and the data are not aligned with their results using B16F10 cells. They can only see an impact of delayed DLN radiation on ICI response when addressing the abscopal effect but not the primary lesions.
- While anti-CTLA-4 is now an accepted treatment modality for melanoma and other cancers, its high dependence on heightening the DC-CD8 T cell synapse makes the results not unexpected. That said, the rationale for the use of anti-CTLA-4 vs anti-PD-1/PDL-1, the most widely used ICIs, should be included.

We agree that an investigation of our hypotheses in additional tumor models and using additional clinically relevant therapies (e.g. anti-PD-1) would increase the relevance of our findings for human cancer. The suggested immune evasive and ICI resistant tumor model such as B16F10 (in addition to the B16F10-Luc model in the original manuscript) is particularly interesting, which is why we have included it into our revision. However, it should be kept in mind that no matter how many different murine models are used to investigate a clinically relevant question, the findings are limited and primarily hypothesis-generated due to the inherent limitations of murine models for human cancer, which cannot be fully overcome. Nevertheless, we now utilized several different experimental model systems. As detailed in the section “Adjuvant draining lymph node irradiation improves

regional lymph node control, mitigates the growth of a distant (non-irradiated) tumor and allows for the induction of long-lasting tumor-specific immunity” (pages 13-16), we compared concomitant and adjuvant treatment regimens using an additional unilateral tumor model (MC38 murine colon carcinoma) and two bilateral tumor models (B16F10 wild type murine melanoma and MC38 murine colon carcinoma), a clinically relevant fractionation schedule (8 Gy x 3) and an alternative widely used ICI (anti-PD-1) (Fig. 6, D to E and Fig. S2, B to N). All data obtained from these additional studies are aligned with our results using B16F10-Luc cells and further extend our conclusions of the benefit of delayed draining lymph node irradiation for the primary tumor. In addition to the corroboration of the improvement of the primary (irradiated) tumor response in the B16F10 wild type tumor model, we have now also observed a strong increase in the occurrence of the distant (non-irradiated, “abscopal”) tumor response in both the immunologically cold B16F10 wild type tumor and another immunologically hot MC38 tumor model, using a-CTLA-4 and/or a-PD-1.

- The potential role of the CCR7-CCL19/CCL21 homing axis as the basis of their observations is correlative in nature, based on decreased levels of the corresponding ligands after DLN irradiation, rather than mechanistic.

Although highly novel and of high potential significance, we agree with the view that our results are correlative with respect to the CCR7-CCL19/CCL21 homing axis. Therefore, following also the suggestions by other reviewers, we have established a collaboration with an expert in the field and performed detailed and extensive series of experiments - to the best of our knowledge for the first time - addressing the importance of the CCR7-CCL19/CCL21 homing axis and the homing chemokine-producing stromal cells of the lymph nodes in response to irradiation. Our findings are presented in Fig. 8, Fig. 9, Fig. 10 and Fig. S4 and in the corresponding sections of the revised manuscript (pages 19-24) and their implications are further interpreted in the revised “Discussion” section. As mentioned also in the “Discussion”, it should be emphasized that additional mechanism-oriented studies will be important but challenging, as any kind of manipulations within the CCR7-CCL19/CCL21 axis, such as using knockout mice or neutralizing antibodies, will undoubtedly completely abrogate immune cell trafficking and thus the development of anti-tumor immunity, given the well-established role of the axis in the immune system (for both T cell and cDC1 homing). Therefore, we believe that such and similar experiments would not contribute to our understanding of the mechanistic link between lymph node IR and the ensuing immunological dysfunction. In our ongoing project, which exceeds the scope of the current study, we are focusing on the development of novel *in vitro*, *ex vivo* and *in vivo* model systems which will allow us to perform more informative mechanism-oriented studies.

Reviewer #2 (Remarks to the Author): with expertise in cancer radiotherapy, immunotherapy

The manuscript by Telarovic et al deals with the important question of the consequences of elective nodal irradiation in the efficacy of radiation/immunotherapy combinations. One strength of this work is the use of a very precise murine model of image-guided lymph node irradiation in metastatic melanoma. However, the main findings (detrimental effect of concomitant DLN irradiation in immune activation and therefore the efficacy of radio-immunotherapy approaches, that can be avoided by delaying DLN irradiation) have long been a concern in the field and have recently been formally proven in other preclinical studies, in particular Marciscano et al 2018 and Darragh et al 2022. Similarly, the role of type I DCs was described in a model of lymphatics-preserving immunotherapy treatment (Saddawi-Konefka). All these studies are known to the authors and cited in the references.

We thank the reviewer for recognizing the clinical relevance of lymph node irradiation in the context of radiotherapy-immunotherapy combinations. We would like to point out, however, that unlike the previous studies, which focus on elective nodal irradiation, our own study does not address the setting of in which lymph node irradiation is elective, but rather focuses primarily on a large subset of patients with nodal involvement or with a high risk of microscopic involvement, whereby lymphatic sparing is not an option and the lymph nodes must be irradiated. Furthermore, having outlined our approach and discussed it with leading radiation oncologists in the field, we believe that adjuvant draining lymph node irradiation needs to be proven as a surgery-independent therapeutic approach compared with neoadjuvant and concomitant draining lymph node irradiation. The investigation and successful demonstration of delayed draining lymph node irradiation, as outlined in our manuscript, is therefore novel and of general interest. We also believe that the extensive investigation of the CCR7-CCL19/CCL21 homing axis and of the stromal cell compartment of the lymph node, performed as part of our revision, strengthens, and strongly increases the novelty of our work.

We would like to extend our gratitude for the remaining helpful comments and suggestions, which we have carefully considered and addressed in our revised manuscript, as detailed below.

Other issues are:

- Local vs regional recurrence vs distant or true metastatic growth are poorly differentiated. For example, despite claiming the use of a metastatic melanoma model, most results are related to growth of the primary tumor. Fig 4H-J introduces a two-tumor (MC38) model and reports survival advantage for the adjuvant DLN IR regimen, but it is unclear whether this is due to primary or abscopal tumor growth delay. Fig 6B shows DLN tumor burden, which is referred to as “metastatic” or “regional” in different sections. This could be an important result, however a single experiment is reported and the adjuvant IR group seems to have less mice than the other groups, therefore statistical significance is unclear.

We apologize for the inconsistency and the lack of clarity with regards to the local vs regional recurrence vs distant or true metastatic growth and thank the reviewer for pointing it out. We have now implemented changes through the manuscript and the figures. To present our data as rigorously as possible from a technical point of view, we now describe lymph node positivity for tumor cells as “presence of tumor cells in the DLNs” instead of “metastatic DLN burden” (see revised Fig. 1C and Fig. 6B). Furthermore, we have more clearly depicted and described our endpoints, i.e. primary (irradiated), secondary (abscopal, non-irradiated) and cumulative (primary + secondary tumor at a given timepoint) tumor volume, time to reach a cumulative volume of 1000 mm³, best response (nadir) value for the primary tumor <50% of the maximal primary tumor volume etc. (see Fig. 6, D to E; Fig. S2, G to N; “Statistical analysis” subsection of the “Methods”, pages 38-39).

We appreciate the point regarding the sample size in Fig. 6B and have therefore increased the sample size by performing additional experiments. The number of mice per group is now balanced between the groups and the results clearly support our hypothesis (see the revised Fig. 6B and page 14 of the “Results” section).

- Some legends are confusing. Eg in Fig 3 “Sham IR” should be renamed “No IR” if mice received no radiation.

We apologize for the confusion stemming from this oversight. Our control group is correctly referred to as “Sham IR” throughout the manuscript, as the mice within this group were indeed sham-irradiated (i.e. underwent the same treatment as the irradiated mice, incl. the anesthesia and a planning CT, but no therapeutic irradiation). Figure legends were corrected to clarify.

- Authors should focus on what could be novel in this study compared to previous publications. The CCL19 data seem to be novel but limited and of unclear translational value, although could be further developed.

We thank the reviewer for recognizing the novelty and the potential significance of the data related to the CCR7-CCL19/CCL21 homing axis. We agree that the findings reported in our initial submission were limited and preliminary. Therefore, following also the suggestions by other reviewers, we have established a collaboration with an expert in the field and performed a detailed and extensive series of experiments addressing the importance of the CCR7-CCL19/CCL21 homing axis and the homing chemokine-producing stromal cells of the lymph nodes. Our findings are presented in Fig. 8, Fig. 9, Fig. 10 and Fig. S4 and in the corresponding sections of the revised manuscript (pages 19-24) and their implications are further interpreted in the revised “Discussion” section. However, as mentioned also in the “Discussion”, it should be emphasized that additional mechanism-oriented studies will be important but challenging, as any kind of manipulations within the CCR7-CCL19/CCL21 axis, such as using knockout mice or neutralizing antibodies, will undoubtedly completely abrogate immune cell trafficking and thus the development of anti-tumor immunity, given the

well-established role of the axis in the immune system (for both T cell and cDC1 homing). Therefore, we believe that such and similar experiments would not contribute to our understanding of the mechanistic link between lymph node IR and the ensuing immunological dysfunction. In an ongoing project, which exceeds the scope of the current study, we are focusing on the development of novel *in vitro*, *ex vivo* and *in vivo* model systems which will allow us to perform more informative mechanism-oriented studies.

- One interesting finding is that in contrast to radio-immunotherapy, the response to radiation alone was not affected by concomitant DLN IR. This is interesting because the “in situ vaccine” concept for IR is commonly accepted in the field. Lack of effects on IR efficacy by concomitant DLN IR could suggest a lack of involvement of the immune system for local response to IR, or that local immunity was sufficient for primary tumor responses. Are T cells required for efficacy? In either case the potential consequence would be that delaying DLN irradiation could be required for immune-radiotherapy but not for radiation alone. Authors could test whether this is also the case in other models and/or radiotherapy regimes.

This is a highly intriguing and important point, and yet often overlooked in the studies investigating the interaction between radiotherapy and anti-tumor immunity. We have considered and even previously investigated the issue in a different tumor model (MC38 murine colorectal carcinoma model; please refer to Telarovic et al., *Neoplasia*, 2022; <https://doi.org/10.1016/j.neo.2022.100812>). In this previous study, we have performed an extensive investigation of the tumor response to different radiotherapy regimens, with or without draining lymph node irradiation, and following different levels of radiation-induced lymphopenia. The experiments were performed in the setting of radiotherapy alone and in the setting of radioimmunotherapy. In line with the B16F10-Luc-related findings from the current study, we were only able to detect an effect of lymph node irradiation in the setting of radioimmunotherapy, but not using radiotherapy alone. We therefore conclude that, as the reviewer suggests, our previous publication together with the current study indeed implicate that delaying lymph node irradiation could only be relevant in the setting of radiotherapy-immunotherapy combinations. Although important and interesting, further investigation of this topic falls out of the scope of the current manuscript and is being investigated as part of our other projects.

Reviewer #3 (Remarks to the Author): with expertise in cancer radiotherapy, immunotherapy

In the manuscript “Delayed tumor draining lymph node irradiation as a new paradigm for radiotherapy-immunotherapy combinations in metastatic disease” (#NCOMMS-23-36401), Telarovic and colleagues investigated whether irradiation of the tumor draining lymph node (TDLN) in neoadjuvant, concomitant or in adjuvant setting modified the response of radiation (RT) with immune checkpoint blocker (ICB). The rationale behind this study relies on the fact that clinical trial testing RT+ICB failed in clinic to generate systemic anti-tumor immunity presumably due to the irradiation of TDLN.

While in preclinical models, it is doable to spare TDLN, in clinic this practice can be questioned due to the metastatic spread in the lymph nodes. Therefore, in metastatic setting, the irradiation of TDLN might be mandatory. Therefore, authors investigated how irradiation of TDLN at different time-point affect RT-induced anti-tumor immunity in the context of ICB and concluded that delayed (adjuvant) irradiation of the TDLN overcomes the detrimental effects of concomitant irradiation of TDLN to induce T-cell mediated responses.

The message of this article is important, novel and within the scope of Nature Communications. I have NO concerns about image manipulation or data fabrication. However, some of the conclusions are not supported by the dataset, and hence must be appropriately revised (or additional experiments performed in this respect). Alongside, some minor issues emerged that need to be addressed, as detailed here below.

We sincerely thank the reviewer for the positive feedback and especially for recognizing the high translational relevance of our study. Furthermore, we appreciate the insightful comments which have helped us in increasing the impact of our work. As suggested, we have performed additional experiments to support and extend our conclusions, and addressed the additional minor issues raised by the reviewer.

Comments:

- Data obtained with the MC38 model is less convincing than the in vivo data obtained with the B16F10-Luciferase model. It is well known that Luciferase is highly immunogenic (at least in preclinical models of breast cancer; PMID: 38798322). Therefore, it is possible that the difference in immunogenicity between these two models is only due to the addition of Luciferase. This potential confounder is also supported by the fact that rechallenge experiment worked in B16F10Luc mice but not in MC38 ones. Authors must repeat in vivo experiments using the B16F10 wild-type model to ensure that their data are not a consequence of the immunogenicity from Luc.

We agree additional tumor models (especially characterized by different levels of immunogenicity) would increase the translational relevance of our findings for human cancer. However, it should be kept in mind that no matter how many different murine models are used to investigate a clinically relevant question, the findings are limited and primarily hypothesis-generated due to the inherent limitations of murine models for human cancer, which cannot be fully overcome. Nevertheless, we found the suggestion to use B16F10 wild

type counterpart of our Luc-expressing model very interesting and of high importance, and therefore repeated the in vivo efficacy experiments using this model. Furthermore, we performed additional experiments in the MC38 model to strengthen and corroborate our hypotheses also in this tumor model. As detailed in the section “Adjuvant draining lymph node irradiation improves regional lymph node control, mitigates the growth of a distant (non-irradiated) tumor and allows for the induction of long-lasting tumor-specific immunity” (pages 13-16), we compared concomitant and adjuvant treatment regimens using a unilateral tumor model (MC38 murine colon carcinoma) and two bilateral tumor models (B16F10 wild type murine melanoma and MC38 murine colon carcinoma), a clinically relevant fractionation schedule (8 Gy x 3) and an alternative widely used ICI (anti-PD-1) (Fig. 6, D to E and Fig. S2, B to N). All data obtained from these additional studies are aligned with our results using B16F10-Luc cells and further extend our conclusions of the benefit of delayed draining lymph node irradiation for the primary tumor. In addition to the corroboration of the improvement of the primary (irradiated) tumor response in the B16F10 wild type tumor model, we have now also observed a strong increase in the occurrence of the distant (non-irradiated, “abscopal”) tumor response in both the immunologically cold B16F10 wild type tumor and another immunologically hot MC38 tumor model, using a-CTLA-4 and/or a-PD-1.

With regards to the rechallenge experiment, we believe there has been a misunderstanding due to an insufficiently clear explanation of the experiment from our side. We have now rearranged the corresponding subfigures for clarity (Fig. 6C and Fig. S2A). All rechallenged mice were originally carrying B16F10-Luc tumors. In order to test for the tumor antigen-specificity of the memory response, we rechallenged some of these cured mice with an antigenically unrelated tumor (MC38). Therefore, the growth of MC38 tumor in those mice was fully expected and in line with the hypothesis that the memory response which developed in the cured mice is indeed specific for their original B16F10-Luc tumor, and consequently did not affect the take rate of a foreign tumor.

- Along similar lines, authors rechallenged B16F10-Luc surviving mice with fresh inoculum of B16F10-Luc but it would have been more convincing to rechallenge with B16F10-wild type.

We agree that this would be an interesting experiment, especially given the raising recognition of the importance of tumor heterogeneity in the context of (radio)immunotherapy. However, given the scope of this question and its implications, we believe it makes more sense to perform this and similar investigations as part of a separate project focused on strategies to induce and optimize epitope spreading (see e.g. Brossart et al., 2020, Clin Cancer Res).

- Authors mentioned that they rechallenged mice from Fig.3 and Fig.4 at day 60. However, sometimes, tumor regrowth occurs around day 80-90. Can authors justify why they decided to rechallenge at that time point and not wait an additional 30-40 days prior to rechallenge to ensure no tumor regrowth?

Unfortunately, due to the strict animal welfare laws, keeping these experimental animals for a longer period of time was not allowed within the framework of the animal license under which this study was performed. However, in our other projects, using similar tumor models, we have never observed tumor regrowth occurring at such a late timepoint. Furthermore, we do not believe that the (highly improbable) tumor regrowth would affect our conclusions or detract for the significance of our findings in this context.

- T cells depletion is needed to confirm the role of T cells in their system.

We have considered this strategy. However, the role of T cells in radiation-induced anti-tumor immunity and in the response to (radio)immunotherapy is very well known and investigated. Moreover, previous studies investigating elective lymph node irradiation have confirmed the absence of the beneficial effect of combining radiotherapy and immunotherapy following CD4⁺ and/or CD8⁺ T cell depletion (Darragh, 2022, Nat Commun). Therefore, we do not believe that T cell depletion would lead to a significant gain of novel knowledge in our study.

- Fig 1G: it seems that gH2AX staining is increase in the non-tumor draining lymph nodes (NTDLN) in the conditions Tumor IR and Tumor IR+DLN IR. Can authors indicate which (NTDLN) they analyzed and provide semi-quantification of their staining? Also, I could not find how many mice neither how many picture were taken to assess the gH2AX staining. Please clarify.

gH2AX staining in this study was performed only as a (bio)dosimetric illustration of our high-precision, volume-oriented small animal radiotherapy treatment approach, for which the framework was previously described and verified by us (Telarovic et al., 2020, Phys Med Biol; <https://doi.org/10.1088/1361-6560/abbb75>). As such, gH2AX staining was not performed with the aim of quantifying and/or statistically analyzing it.

With regards to the apparent increase in the gH2AX positive cells in the non-draining lymph nodes in the “Tumor IR” and the “Tumor IR + DLN IR” groups which the reviewer mentions, it could be speculated that irradiation of the blood passing through the irradiation field (and thus of circulating lymphocytes) leads to a general increase in occurrence of double stranded breaks (i.e. of gH2AX positivity) in the lymphocytes, which rapidly recirculate and reach different organs, including the non-draining lymph nodes. The apparent increase in the gH2AX positivity also in the draining lymph nodes of the “Tumor IR” group in comparison to the “CT only” group would support this speculation. Furthermore, from our previous study on the influence of radiotherapy treatment volume on radiation-induced lymphopenia, we know that even smaller doses, delivered using very small, tumor-only directed radiation fields can induce significant lymphopenia (Telarovic et al., Neoplasia, 2022; <https://doi.org/10.1016/j.neo.2022.100812>). Thus, such a transient, but systemic consequence of radiotherapy on the circulating lymphocytes cannot be excluded. However, the investigation of such phenomena, while intriguing, is out of the scope of this study and does not influence its conclusions.

- Authors mentioned that they irradiated by 15Gy. It is not clear if this is the total dose meaning that the mice received 5Gy in the tumor + 5Gy in the TDLN (3x5Gy) or if the mice received 15Gyx3. Please clarify.

We apologize for the confusion. In the experiments using a single high-dose approach, each target received a total single dose of 15 Gy. In case of concomitant draining lymph node irradiation, the mouse was thus irradiated using 3 separate fields (field 1: tumor, field 2: inguinal draining lymph node, field 3: axillary and brachial draining lymph nodes), with each field delivering 15 Gy to its target. In the experiments using fractionated (8Gyx3) radiotherapy, each target received 3 fractions delivering 8 Gy per fraction. We added the dosing information to the illustrations of the experimental workflow and adapted the figure legends to clarify.

- The radiation dose regimen used is not clinically relevant and is not justify in their manuscript. Authors not only should justify their choice but should consider running an experiment with a clinically relevant dose fractionation.

We appreciate the point raised by the reviewer. As detailed in the section “Adjuvant draining lymph node irradiation improves regional lymph node control, mitigates the growth of a distant (non-irradiated) tumor and allows for the induction of long-lasting tumor-specific immunity” (pages 13-16), we compared concomitant and adjuvant treatment regimens using additional tumor models and a clinically relevant immunomodulatory hypofractionation schedule (8 Gy x 3) (Fig. 6, D to E and Fig. S2, B to N). All data obtained from these additional studies are aligned with the studies using single high dose irradiation in a B16F10-Luc tumor model. In addition to the corroboration of the improvement of the primary (irradiated) tumor response in the B16F10 wild type tumor model treated using hypofractionation, we have also observed a strong increase in the occurrence of the distant (non-irradiated, “abscopal”) tumor response in both the immunologically cold B16F10 wild type tumor and another immunologically hot MC38 tumor model, using a-CTLA-4 and/or a-PD-1.

- Authors have performed flowcytometry to assess the T cells compartment throughout their study. However, the definition of Tregs is incomplete in mice. This cell subtype should be defined as live CD45⁺ FoxP3⁺CD25⁺ which is not what is presented in this manuscript (live CD45⁺ FoxP3⁺).

The necessity of CD25 in the definition of suppressive regulatory T cells in mice is debatable, especially considering the existence of CD25⁻ and CD25^{low} FOXP3⁺ regulatory T cells (reviewed in e.g. Li et al., 2015, Cell Mol Immunol). Nevertheless, in order to ensure the accuracy of our findings and the validity of our conclusions, we compared regulatory T cells proportions obtained by defining regulatory T cells as CD4⁺FOXP3⁺ (as done throughout our study) versus CD4⁺CD25⁺FOXP3⁺ (as suggested by the reviewer). In a mixed population of 18 mice treated with the treatment regimens used throughout the study, we observed no differences in the proportions of regulatory T cells related to the presence of CD25 in the

definition of this subset (Fig. P1). Therefore, we conclude with high probability that the absence of CD25 in the definition of regulatory T cells in our study does not weaken our data or influence our conclusions.

Figure P1. Regulatory T cells: CD4⁺FOXP3 versus CD4⁺CD25⁺FOXP3⁺. (A) The proportion of regulatory T cells within the CD4 compartment without (left) or with (right) using CD25⁺ in gating. (B) Representative flow cytometry plots.

- Authors should provide a more in depth characteristic of their T cell phenotype than just only look at CD8 expression. Meaning: do they express activation markers? How about PD-1 expression?

We thank the reviewer for this suggestion, which resulted in the several intriguing observations and further solidified our conclusions. New data with an extended phenotypical characterization of both tumor- and lymph node-residing T cells is presented in Fig. 3, G to H and Fig. S1D (for tumor-infiltrating T cells), and in Fig. 7 (for T cells residing in the draining lymph nodes). The corresponding interpretation can be found in the revised “Results” section (pages 9-10 and page 18).

- The disruption of the CCR7-CCL19/CCL21 with radiation of the TDLN is weak. Only 1 experiment is suggesting this mechanism (Fig. 7). To strengthen their findings, authors should disrupt this axis in the RT+ICB animal (no irradiation of the TDLN) to determine whether the tumor control and recruitment of CD8 T cells in RT+ICB animals is due to the homing of DC1 via CCR7-CCL19/CCL21

Although highly novel and of high potential significance, we agree with the notion on the preliminary nature of our findings related to the CCR7-CCL19/CCL21 homing axis. Therefore, following also the suggestions by other reviewers, we have established a collaboration with an expert in the field and performed a detailed and extensive series of

experiments addressing the importance of the CCR7-CCL19/CCL21 homing axis and the homing chemokine-producing stromal cells of the lymph nodes. Our findings are presented in Fig. 8, Fig. 9, Fig. 10 and Fig. S4 and in the corresponding sections of the revised manuscript (pages 19-24) and their implications are further interpreted in the revised “Discussion” section. However, as mentioned also in the “Discussion”, it should be emphasized that additional mechanism-oriented studies will be important but challenging, as any kind of manipulations within the CCR7-CCL19/CCL21 axis, such as using knockout mice or neutralizing antibodies, will undoubtedly completely abrogate immune cell trafficking and thus the development of anti-tumor immunity, given the well-established role of the axis in the immune system (for both T cell and cDC1 homing). Therefore, we believe that such and similar experiments would not contribute to our understanding of the mechanistic link between lymph node IR and the ensuing immunological dysfunction. In an ongoing project, which exceeds the scope of the current study, we are focusing on the development of novel *in vitro*, *ex vivo* and *in vivo* model systems which will allow us to perform more informative mechanism-oriented studies.

Reviewer #4 (Remarks to the Author): with expertise in cancer radiotherapy, immunotherapy

this is a very interesting topic addressed by this study since the optimal scheme of radio immunotherapy in terms of tumor volume and lymph node irradiation is poorly defined. A better understanding of the interplay between tumor lymph node irradiation and tumor response to immunoradiotherapy should contribute to improve the results of such combinations in the clinic.

The manuscript reads well overall and the experiments are adequately described providing evidence that delayed lymph node irradiation is superior to neo adjuvant or concomitant irradiation when anti CTLA-4 is added to radiotherapy in terms of local control and systemic immunity.

We kindly thank the reviewer for recognizing the high translational relevance of our study and for the positive feedback. Moreover, we thank the reviewer for their helpful suggestions on how to maximize the clinical relevance of our data and for pointing out the aspects which were insufficiently explained in the original submission. We believe that, with the additional experiments and the adaptations in the flow of the manuscript fully, we were able to fully address the reviewer's suggestions and comments, as outlined below.

Some aspects remains to be clarified :

-what is the evidence of nodal tumor involvement dependency in these models? Is tumor lymph node involvement a prerequisite for the benefit of delayed lymph node irradiation?

In our study, we primarily address a clinical situation in which lymph node sparing is **not** possible due to the spread of the disease. We investigate an alternative possible approach in this situation, which allows for the retention of the clinically unavoidable lymph node irradiation, while simultaneously allowing for the development of tumor irradiation-induced anti-tumor immunity. To mimic this clinical situation on the preclinical level, we primarily focused on a B16F10-Luc mouse tumor model with evidence of early nodal involvement (as shown in Fig. 1, B and C), which indeed benefited from delayed lymph node irradiation. However, from the additional experiments performed as part of the revision, we now also demonstrate the benefit of delayed lymph node irradiation in additional tumor models: B16F10 wild type and MC38. We did not evaluate nodal involvement in these tumor models, as this was not a prioritized endpoint for these additional experiments. Based on the previous studies, the B16F10 wild type tumor model (similar to the B16F10-Luc mouse tumor model) can also be considered highly metastatic and nodal involvement at the time of treatment is a possibility. On the other hand, in a study using a similar MC38 tumor model there was no evidence of nodal involvement (Marciscano et al., 2018, Clin Cancer Res). Regardless of the exact nodal status in our experiments, our hypothesis-generating data suggest that delayed lymph node irradiation is a feasible and beneficial strategy in the setting of both curative and elective nodal irradiation, although the latter was not the focus of the current study. Due to the inherent limitations of murine model systems for human cancer, the reviewer's question

of the relevance of nodal involvement (i.e. the evidence of clinical nodal positivity at the time of treatment) will only be fully answered following randomized clinical trials and will highly likely strongly depend on the tumor type and stage, as well as patient characteristics.

-In order to maximize the clinical relevance of these data, is there a way to add an experiment that would be more compatible with the radiotherapy regimens performed into the clinic since it is likely that the single dose radiation schemes will not be applicable to tumors where elective lymph node irradiation is performed (ie fractionated irradiation)

We appreciate the point raised by the reviewer. As detailed in the section “Adjuvant draining lymph node irradiation improves regional lymph node control, mitigates the growth of a distant (non-irradiated) tumor and allows for the induction of long-lasting tumor-specific immunity” (pages 13-16), we compared concomitant and adjuvant treatment regimens using additional tumor models and a clinically relevant immunomodulatory hypofractionation schedule (8 Gy x 3) (Fig. 6, D to E and Fig. S2, B to N). All data obtained from these additional studies are aligned with the studies using single high dose irradiation in a B16F10-Luc tumor model. In addition to the corroboration of the improvement of the primary (irradiated) tumor response in the B16F10 wild type tumor model treated using hypofractionation, we have also observed a strong increase in the occurrence of the distant (non-irradiated, “abscopal”) tumor response in both the immunologically cold B16F10 wild type tumor and another immunologically hot MC38 tumor model, using a-CTLA-4 and/or a-PD-1.

-the identification of the role of CCR7-CCL19/CCL21 axis is very interesting but functional data would strengthen the assumption. Manipulation of chemokines using ko mice and or administration of chemokines or blocking antibodies in the same experiments could contribute to reinforce their statements. the statements in the abstract and the manuscript conclusion should be adapted accordingly.

Although highly novel and of high potential significance, we agree with the notion on the preliminary nature of our findings related to the CCR7-CCL19/CCL21 homing axis and the necessity to perform functional studies. Therefore, following also the suggestions by other reviewers, we have established a collaboration with an expert in the field and performed a detailed and extensive series of experiments addressing the importance of the CCR7-CCL19/CCL21 homing axis and the homing chemokine-producing stromal cells of the lymph nodes. Our findings are presented in Fig. 8, Fig. 9, Fig. 10 and Fig. S4 and in the corresponding sections of the revised manuscript (pages 19-24) and their implications are further interpreted in the revised “Discussion” section. However, as mentioned also in the “Discussion”, it should be emphasized that additional mechanism-oriented studies will be important but challenging, as any kind of manipulations within the CCR7-CCL19/CCL21 axis, such as using knockout mice or neutralizing antibodies, will undoubtedly completely abrogate immune cell trafficking and thus the development of anti-tumor immunity, given the well-established role of the axis in the immune system (for both T cell and cDC1 homing). Therefore, we believe that such and similar experiments would not contribute to our

understanding of the mechanistic link between lymph node IR and the ensuing immunological dysfunction. In an ongoing project, which exceeds the scope of the current study, we are focusing on the development of novel *in vitro*, *ex vivo* and *in vivo* model systems which will allow us to perform more informative mechanism-oriented studies.

The interpretation of the experiments in the body of the manuscript could be streamlined in order to ease the reader's understanding as well as the figures legends.

We appreciate the suggestion and agree that the original presentation of the data was not optimal. With the aim of easing the understanding, we have reorganized the figures and streamlined the flow of the manuscript. We hope the revised version is easier to follow.

REVIEWER COMMENTS

Reviewer #3 (Remarks to the Author):

In the revised manuscript “Delayed tumor draining lymph node irradiation as a new paradigm for radiotherapy-immunotherapy combinations in metastatic disease” (#NCOMMS-23-36401A-Z), Telarovic and colleagues have improved their manuscript to clarify concerns and increase the robustness of their data.

For instance, they have gathered new data in other experimental models that confirm their main findings and change the radiation schedule to use a clinically relevant one.

However, some concerns need to be addressed.

Here are my comments:

- Fig 1G: authors mentioned in their rebuttal letter that gH2AX was quote: “performed as a (bio) dosimetric illustration [...] for which the framework was verified by us. As such, gH2AX staining was not performed with the aim of quantifying”. Authors should clearly mention this aspect in the results section to justify the absence of quantification and mention that it is just for appreciation but not necessarily representative. “n” should be also clearly stated in the text as well as which (inguinal or else) was used for the assay.
- Fig 1G: justification of gH2AX staining in non-irradiated lymph nodes is speculative but should also be added in the manuscript.
- Fig 6E: Abscopal responses are evaluated as cumulative tumor volume. While it is interesting and of value to show individual mice tumor growth curve, the mean of the primary and secondary tumors should be showed and statistical analysis using one-way ANOVA should be performed on each graph and not on the cumulative tumor volume curves. Same comment apply for Figure S2H and S2L
- Fig 9B: There is no statistical significance of CCL19 protein concentration from Day 2 to day 9. Therefore, authors should refrain to conclude that there was a partial recovery at day 9 (line 462 page 21).
- Data pertaining to the CCR7-CCL19/CCL21 axis are still correlative and not causative in nature. There is no mention of CCR7 in their system. While it is appreciated that targeting this axis might hinder the trafficking of all immune cells, authors should perform migration assay in vitro or similar to at least show causation.

- Flow cytometry gating strategy of T cells should be added.

Reviewer #4 (Remarks to the Author):

The manuscript can now be published as is, the additional data on other tumor models, changes and insertions throughout the manuscript are very satisfactory.

Manuscript by Irma Telarovic, Carmen S.M. Yong, Lisa Kurz, Irene Vetrugno, Sabrina Reichl, Alba Sanchez Fernandez, Hung-Wei Cheng, Rona Winkler, Matthias Guckenberger, Anja Kipar, Burkhard Ludewig, and Martin Pruschy titled “Delayed Tumor Draining Lymph Node Irradiation as a New Paradigm for Radiotherapy-Immunotherapy Combinations in Metastatic Disease” (NCOMMS-23-36401)

We thank the reviewers for their assessment of our revised manuscript. We appreciate the additional constructive suggestions. We have addressed the issues kindly pointed out by Reviewer #3.

Please note the following nomenclature used for referring to the figures:

Fig. 1 refers to main Figure 1 of the manuscript.

Fig. S1 refers to Supplementary Figure S1 of the manuscript.

Reviewer #3

In the revised manuscript “Delayed tumor draining lymph node irradiation as a new paradigm for radiotherapy-immunotherapy combinations in metastatic disease” (#NCOMMS-23-36401A-Z), Telarovic and colleagues have improved their manuscript to clarify concerns and increase the robustness of their data. For instance, they have gathered new data in other experimental models that confirm their main findings and change the radiation schedule to use a clinically relevant one. However, some concerns need to be addressed.

We kindly thank the reviewer for the positive feedback and for their previous suggestions which have greatly contributed to the improvement of our original submission. Furthermore, we thank the reviewer for their additional helpful comments. We believe that we were now able to fully address the remaining issues.

Here are my comments:

- Fig 1G: authors mentioned in their rebuttal letter that gH2AX was quote: “performed as a (bio) dosimetric illustration [...] for which the framework was verified by us. As such, gH2AX staining was not performed with the aim of quantifying”. Authors should clearly mention this aspect in the results section to justify the absence of quantification and mention that it is just for appreciation but not necessarily representative. “n” should be also clearly stated in the text as well as which (inguinal or else) was used for the assay.
- Fig 1G: justification of gH2AX staining in non-irradiated lymph nodes is speculative but should also be added in the manuscript.

In the revised “Results” section, we have now clarified the illustrative, rather than quantitative nature of the gH2AX staining shown in Fig. 1G. Furthermore, we explicitly mention that we analyzed 6 lymph nodes from each mouse, and that the figures show sections from the representative inguinal lymph nodes of the three mice treated with the three different treatment approaches. “n=1 mouse per group, 6 lymph nodes per mouse” has been added to the corresponding figure legend.

The speculation regarding the gH2AX staining in the non-irradiated lymph nodes has likewise been added to the “Results” section.

- Fig 6E: Abscopal responses are evaluated as cumulative tumor volume. While it is interesting and of value to show individual mice tumor growth curve, the mean of the primary and secondary tumors should be showed and statistical analysis using one-way ANOVA should be performed on each graph and not on the cumulative tumor volume curves. Same comment apply for Figure S2H and S2L

We have added the mean tumor volume curves for the primary and secondary tumors and performed the additional statistical analyses, as suggested. Please see revised Fig. S2B, Fig.

S3G and Fig. S3L, along with additions to the “Results” section (page 16 and 17). We thank the reviewer for their recommendation, which strengthened the validity of our previous interpretation of the abscopal responses.

- Fig 9B: There is no statistical significance of CCL19 protein concentration from Day 2 to day 9. Therefore, authors should refrain to conclude that there was a partial recovery at day 9 (line 462 page 21).

We thank the reviewer for pointing out this issue. We have removed the mention of a partial recovery and rephrased the text to clarify (page 22).

- Data pertaining to the CCR7-CCL19/CCL21 axis are still correlative and not causative in nature. There is no mention of CCR7 in their system. While it is appreciated that targeting this axis might hinder the trafficking of all immune cells, authors should perform migration assay in vitro or similar to at least show causation.

As recommended, we performed additional experiments to investigate the causal relationship between irradiation, disruption of the axis and the immunological lymph node dysfunction. Please see: Fig. 9, E to G and Fig. S5D; “Results” section, pages 23-24; “Discussion” section, pages 29-30. We believe that these additional experiments, in particular the recommended migration assay, have strongly solidified our findings and thank the reviewer for their valuable suggestion.

- Flow cytometry gating strategy of T cells should be added.

We apologize for the oversight and have now added the missing parts of the gating strategy. Please see revised Fig. S1A for the gating strategy used to characterize T cells in tumors and the revised Fig. S4A for the gating strategy used to characterize T cells in the lymph nodes. All supplementary figures detailing the gating strategies are now also explicitly referenced in the Reporting Summary.

REVIEWERS' COMMENTS

Reviewer #3 (Remarks to the Author):

The authors have cleared my concerns.